# Coordination of cell cycle and morphogenesis during organ formation

Jeffrey Matthew, Vishakha Vishwakarma, Thao Phuong Le, Ryan A Agsunod, SeYeon Chung*

Department of Biological Sciences, Louisiana State University, Baton Rouge, United States

**Abstract** Organ formation requires precise regulation of cell cycle and morphogenetic events. Using the *Drosophila* embryonic salivary gland (SG) as a model, we uncover the role of the SP1/KLF transcription factor Huckebein (Hkb) in coordinating cell cycle regulation and morphogenesis. The *hkb* mutant SG exhibits defects in invagination positioning and organ size due to the abnormal death of SG cells. Normal SG development involves distal-to-proximal progression of endoreplication (endocycle), whereas *hkb* mutant SG cells undergo abnormal cell division, leading to cell death. Hkb represses the expression of key cell cycle and pro-apoptotic genes in the SG. Knockdown of *cyclin E* or *cyclin-dependent kinase 1,* or overexpression of *fizzy-related* rescues most of the morphogenetic defects observed in the *hkb* mutant SG. These results indicate that Hkb plays a critical role in controlling endoreplication by regulating the transcription of key cell cycle effectors to ensure proper organ formation.

## Editor's evaluation

The important paper provides compelling evidence that the gene hkb controls the development of the salivary gland of *Drosophila* by controlling a cell cycle and cell death. The work extends and corrects previous findings on the role of this transcription factor and will be important for scientists interested in organogenesis and in particular the coordination of cell cycle and cell death.

*For correspondence:
seyeonchung@lsu.edu

**Competing interest:** The authors declare that no competing interests exist.

## Introduction

The formation of tubular organs, such as the lungs and kidneys, begins with the folding of flat epithelial tissues into functional three-dimensional (3D) tubular structures. In humans, errors in tube formation can result in congenital defects such as spina bifida and lung hypoplasia (*Wilde et al., 2014*). Studies in both vertebrates and invertebrates have revealed key morphogenetic mechanisms underlying epithelial tube formation during development (*Andrew and Ewald, 2010*; *Chung et al., 2014*; *Herriges and Morrisey, 2014*; *Girdler and Röper, 2014*; *Nikolopoulou et al., 2017*; *Spina and Cowin, 2021*; *Pradhan et al., 2023*). While proper cell shape changes and cell rearrangement are critical, precise regulation of cell division and cell death is also essential for tissue morphogenesis (*Huh et al., 2004*; *Mao et al., 2013*; *LeGoff and Lecuit, 2015*; *Godard and Heisenberg, 2019*). Notably, the interplay between cell division and cell death provides a homeostatic control system necessary for proper organ formation. Perturbations at any stage of the cell cycle can have detrimental effects on cell viability, leading to cell death, and activation of apoptosis can trigger compensatory proliferation in surrounding cells (*Ryoo et al., 2004*). In addition to their role in establishing and maintaining the correct number of cells in tissues and organs, cell division and cell death generate biophysical forces through actomyosin reorganization that help shape the final tissue morphology. For example, during *Drosophila* tracheal invagination, mitotic cell rounding drives the internalization

of the tracheal primordium (**Kondo and Hayashi, 2013**). During dorsal closure and leg folding in *Drosophila* (**Toyama et al., 2008**; **Monier et al., 2015**) and neural closure in mammals (**Massa et al., 2009**; **Yamaguchi et al., 2011**), apoptotic cells generate forces at both local and tissue scales that affect tissue remodeling. Therefore, to fully understand the mechanisms of organ formation, it is critical to know how the cell cycle, cell death, and morphogenesis are coordinated during epithelial tube formation.

The *Drosophila* embryonic SG is a premier model for studying the mechanisms of tube formation from a flat epithelial sheet (**Chung et al., 2014**; **Sidor and Röper, 2016**). The *Drosophila* embryo contains two SGs, each consisting of 140–150 cells, which initially reside on the embryo surface and invaginate to form single-layered, elongated tubes (**Figure 1A**). Once SG cells are specified, they do not divide or die. Cell death is actively inhibited during SG morphogenesis. The SG-upregulated transcription factors Fork head (Fkh), Senseless, and Sage inhibit apoptosis by downregulating proapoptotic genes, such as *head involution defective* (*hid*) and *reaper* (*rpr*) (**Myat and Andrew, 2000a**; **Chandrasekaran and Beckendorf, 2003**; **Fox et al., 2013**). Instead of cell division, SG cells undergo endoreplication (endocycle), a variant of the cell cycle that involves DNA replication without subsequent cell division forming polyploid cells, upon invagination at stage 11 (**Follette et al., 1998**). SG cells that have completed one round of endoreplication arrest the cell cycle in G2 until the larval stage (**Smith and Orr-Weaver, 1991**; **Chandrasekaran and Beckendorf, 2005**). Endoreplication, which is more common in invertebrates and plants, was thought to be a rare cell cycle variation in mammals. However, recent studies have revealed endoreplication in various mammalian tissues, including humans, and endoreplication is emerging as a potential developmental timer (**Gandarillas et al., 2018**). Like the conventional cell cycle, endoreplication is regulated by key cell cycle genes such as cyclins and cyclin-dependent kinases (CDKs). In *Drosophila* embryogenesis, re-entry into S phase during endoreplication is regulated by the cyclin E-Cdk2 complex through the E2F proteins (**Duronio et al., 1995**; **Follette et al., 1998**). Because SG morphogenesis occurs in the absence of cell division and cell death, studies of SG morphogenesis have primarily focused on the mechanisms underlying cell shape changes, cell intercalation, and collective cell migration (**Pirraglia et al., 2010**; **Xu et al., 2011**; **Röper, 2012**; **Ismat et al., 2013**; **Booth et al., 2014**; **Chung et al., 2014**; **Chung et al., 2017**; **Sanchez-Corrales et al., 2018**; **Sánchez-Corrales et al., 2021**; **Le and Chung, 2021**; **Vishwakarma et al., 2022**). Little is known about the potential impact of regulating endoreplication and cell death during SG morphogenesis.

Invagination is the first step in SG morphogenesis that is critical for organ architecture, and studies in recent years have revealed key underlying molecular and cellular mechanisms (**Chung et al., 2017**; **Sanchez-Corrales et al., 2018**; **Sidor et al., 2020**; **Sánchez-Corrales et al., 2021**; **Le and Chung, 2021**; **Vishwakarma et al., 2022**). SG invagination occurs in a stereotypical manner, with two predominant cell behaviors occurring in distinct spatial domains within the placode relative to the invagination pit forming in the dorsal/posterior region: clustered apical constriction near the pit (**Figure 1A**) and cell intercalation far from the pit. Two early-expressed SG-upregulated transcription factors, Fkh and Hkb, play critical roles in SG invagination. Fkh, a winged-helix transcription factor that regulates more than two-thirds of the known SG genes, is essential for SG invagination; SG cells fail to invaginate in *fkh* mutants in which apoptosis is inhibited (**Myat and Andrew, 2000a**; **Maruyama et al., 2011**). Hkb, an SP1/KLF transcription factor, is crucial for invagination positioning; SG cells invaginate in the central region of the placode in *hkb* mutants (**Myat and Andrew, 2000b**). A recent study proposed that the early patterning of Fkh and Hkb and their downstream effectors ensures positionally fixed cell behaviors with respect to invagination position (**Sánchez-Corrales et al., 2021**). For example, a key downstream target of Fkh, Folded gastrulation (Fog), signals through its G protein-coupled receptor to activate Rho kinase and myosin to regulate apical constriction in the dorsal/posterior region of the SG placode (**Chung et al., 2017**; **Vishwakarma et al., 2022**). Despite the intriguing phenotype of *hkb* mutants, the mechanism of how Hkb regulates SG invagination positioning remains largely unknown. Only two potential downstream targets of Hkb involved in SG formation have been identified: *crumbs* (*crb*), encoding a key apical membrane protein, and *klarsicht* (*klar*), encoding a Nesprin family protein that mediates dynein-dependent organelle trafficking (**Myat and Andrew, 2002**). To fully understand the mechanism of how Hkb regulates invagination position, it is critical to reveal the precise cellular mechanisms underlying the centered invagination phenotype and identify Hkb downstream targets regulating the process.

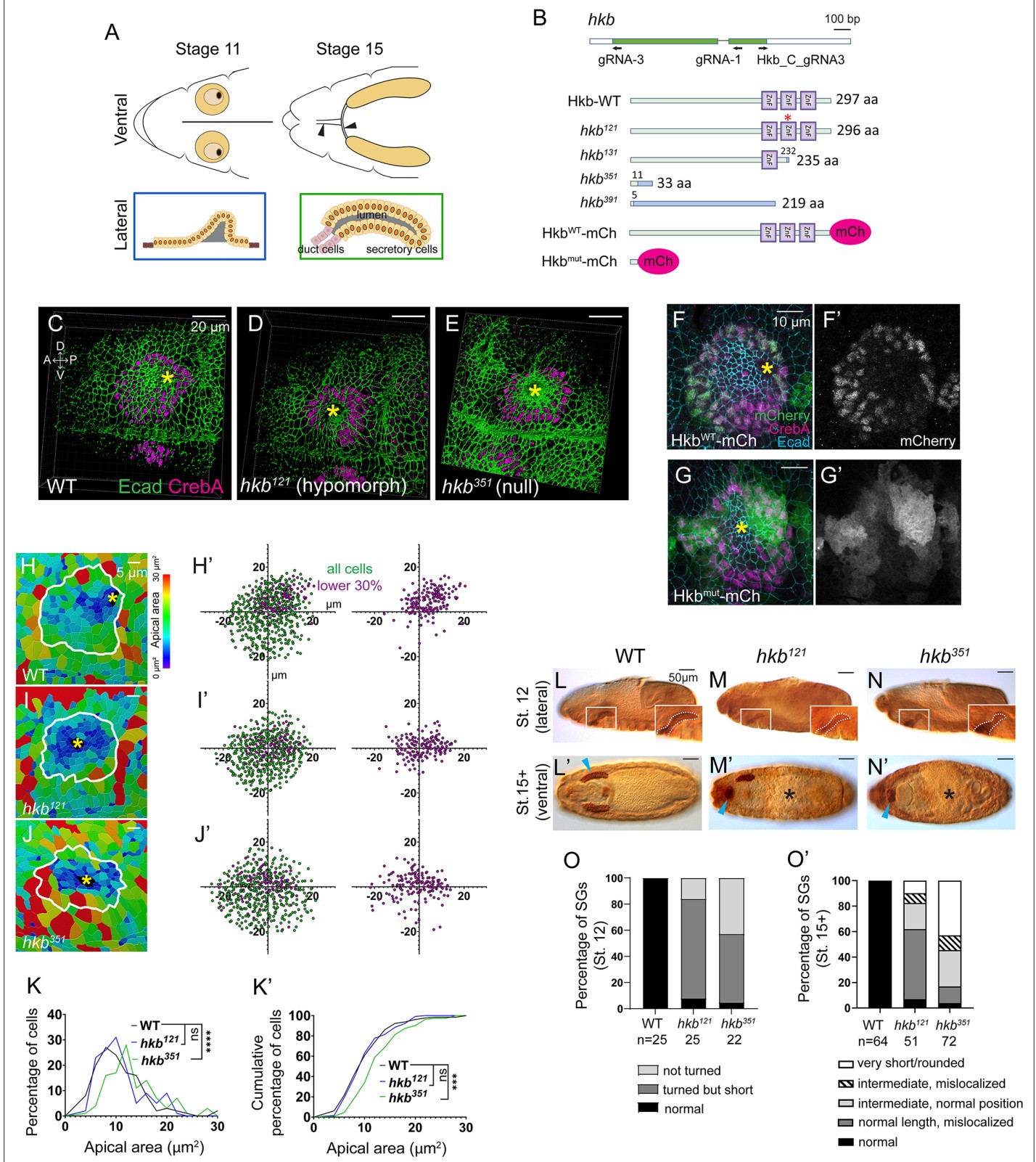

**Figure 1.** Generation of new *huckebein* (*hkb*) alleles and fluorescent knock-in lines reveals its function and tissue-specific localization during salivary gland (SG) morphogenesis. (**A**) Top cartoon diagram: Anterior region of stage 11 and 15 *Drosophila* embryos. SGs shown in yellow. Brown area, clustered apical constriction of SG cells during invagination. Black, invagination pit. Arrowheads, salivary duct. Bottom cartoon diagram: Lateral view of SG at stages 11 and 15. Orange circles, SG cell nuclei. Gray, SG lumen. (**B**) Top cartoon diagram: *hkb* transcript displayed with coding exon in

*Figure 1 continued on next page*

*Figure 1 continued*

green and gRNAs used for *hkb* alleles and mCh knock-in lines. Bottom: Hkb protein structures produced in each allele and knock-in lines. In *hkb[121]*, two amino acids (N237 and E238) are substituted with K237 (asterisk). In *hkb[351]* and *hkb[391]*, a short N-terminal region of the Hkb protein (green) is followed by random amino acid sequences (blue) due to a frameshift. *hkb* mRNA with the premature stop codon in *hkb[351]* might be degraded by nonsense-mediated mRNA decay. ZnF, zinc finger domain. (**C–E**) Three-dimensional (3D) reconstruction of the confocal images of invaginating SGs immunolabeled with E-Cadherin (Ecad) and CrebA. Asterisks, invagination pit. (**F-G′**) Confocal images of stage 11 SGs immunostained for mCh (green), CrebA (magenta), and Ecad (cyan). (**H–J**) Heat maps showing apical area distribution in invaginating SGs. White lines, SG boundary determined by SG-specific CrebA signals. Yellow asterisks, invagination pits. (**H′-J′**) Scatter plots showing the relative position of cells in the SG placode. X and Y axes represent the distance along the A/P and D/V axes, respectively, from the center of the placode. Cells in the bottom 30% of the apical area (magenta) and all cells (green) are plotted (n=5 SGs from different embryos (WT, 572 cells; *hkb[121]*, 505 cells; *hkb[351]*, 498 cells)). (**K, K′**) Quantification of the apical area distribution in the SG cells of H-J. Mann-Whitney U test (percentage of cells) and Kolmogorov-Smirnov test (cumulative percentage of cells). **p<0.01; ***p<0.001; ****p<0.0001. (**L-N′**) Embryos immunostained for CrebA, with magnified views (insets) showing SGs outlined by white dashed lines. Cyan arrowheads, late-stage SGs. Asterisks, defective gut morphology. (**O, O′**) Quantification of the phenotypic distribution of SGs. n, number of embryos. Statistical significance was determined using two-way ANOVA (p<0.05).

The online version of this article includes the following figure supplement(s) for figure 1:

**Figure supplement 1.** Indels of the CRISPR/Cas9 *huckebein* (*hkb*) alleles and predicted protein sequences for each allele.

**Figure supplement 2.** Hkb[wt]-mCh and Hkb[mut]-mCh signals during embryonic development.

**Figure supplement 3.** Quantification of apical areas in the pre-invaginating salivary gland (SG) of wild-type (WT) and *huckebein* (*hkb*) mutant embryos.

**Figure supplement 4.** Salivary gland (SG) phenotypes in different *huckebein* (*hkb*) alleles.

**Figure supplement 5.** *huckebein* (*hkb*) mutants show short salivary duct phenotypes with reduced cell numbers and cell intercalation defects.

In this study, we reveal the distal-to-proximal progression of endoreplication during SG morphogenesis and show that Hkb plays a critical role in this process. Loss of *hkb* results in abnormal cell division in the SG leading to cell death. Subsequently, abnormal cell death leads to malposition of the SG invagination and the formation of a smaller organ. Using modENCODE Hkb ChIP-Seq targets and genetic interaction analyses, we show that Hkb represses the transcription of key cell cycle genes in the SG to control cell cycle progression and ensure cell survival and proper SG morphogenesis.

## Results

### Creation of the molecularly defined *hkb* mutant alleles and fluorescent knock-in lines

To fully characterize the role of Hkb during SG morphogenesis, we generated two nulls (*hkb[351]*, *hkb[391]*) and two hypomorphic alleles (*hkb[121]*, *hkb[131]*) using CRISPR/Cas9 (*Figure 1B*; Details of indels and expected protein sequences were described in *Figure 1—figure supplement 1*). All alleles were embryonic lethal in homozygous embryos and transheterozygotes with a deficiency line deleting the *hkb* locus, suggesting that all three zinc-finger (ZnF) domains are required for embryonic development and survival. Immunostaining with CrebA, a nuclear marker for secretory cells of the SG, and E-Cadherin (Ecad), an adherens junction marker, showed that wild-type (WT) embryos exhibited SG invagination in the dorsal-posterior region of the placode (*Figure 1C*). However, new *hkb* alleles exhibited a broad, central SG invagination phenotype (*Figure 1D and E*), similar to previous studies with an EMS allele *hkb[2]* and a P-element excision allele *hkb[A321R1]* (*Weigel et al., 1990*; *Gaul and Weigel, 1990*; *Myat and Andrew, 2002*; *De Iaco et al., 2006*; *Sánchez-Corrales et al., 2021*). *hkb[121]* also exhibited the centered invagination phenotype, albeit with only 50% penetrance (*Figure 1D*; n=20), indicating that all three ZnF domains are required for the positioning of the SG invagination.

We also generated two mCherry-tagged transgenic lines, Hkb[WT]-mCh (a functional fusion protein line) and Hkb[mut]-mCh (a mutant knock-in line) using CRISPR-mediated homology-directed repair (*Figure 1B*). The Hkb[WT]-mCh line exhibited nuclear mCh signals in several tissues known to express Hkb, including the anterior and posterior termini of the early embryo, SGs, and a subset of neuroblasts (*Figure 1F and F′* and *Figure 1—figure supplement 2*). In the SG, strong Hkb[WT]-mCh signals were detected at stages 10 and 11, with higher levels in the posterior region of the placode at stage 10 (*Figure 1—figure supplement 2C–F″*), consistent with a previous study (*Sánchez-Corrales et al.,*

*2021*). Weak Hkb^WT-mCh signals were present at stages 12 and 13 and not detected thereafter (*Figure 1—figure supplement 2G–J"*), consistent with the transient expression of *hkb* transcripts in the SG (*Myat and Andrew, 2002*). Unlike the nuclear Hkb^WT-mCh signals, Hkb^mut-mCh signals were distributed throughout the cell (*Figure 1G and G'* and *Figure 1—figure supplement 2*). In homozygous Hkb^mut-mCh and heterozygous Hkb^mut-mCh/+embryos, mCh signals persisted throughout the SG cell until stage 16, likely due to the loss of the Hkb protein degradation elements/signals (*Figure 1—figure supplement 2J–J", T-T"*).

To investigate the effect of *hkb* loss on SG cell behavior during invagination at stage 11, we analyzed the distribution of SG cells undergoing apical constriction in WT and *hkb* mutants using cell segmentation analysis (*Figure 1H–K'* and *Figure 1—figure supplement 3*). Before invagination, WT SG cells with small apical areas clustered in the dorsal-posterior region of the placode, whereas *hkb* mutants exhibited a scattered distribution (*Figure 1—figure supplement 3D–F'*). As invagination progressed, more cells with small apical areas were observed near the invagination pit in WT, whereas these cells were distributed around the wide central pit in *hkb* mutant SGs (*Figure 1H–J'*). This is consistent with a previous study using *hkb²* (*Sánchez-Corrales et al., 2021*), although we observed that only about 30% of the *hkb²* embryos showed the central SG invagination phenotype (n=11). The frequency distribution of cells based on their apical areas showed a significantly lower percentage of SG cells with smaller apical areas in *hkb* mutants compared to WT (*Figure 1K and K'* and *Figure 1—figure supplement 3G, G'*), indicating a defect in apical constriction.

Using CrebA signals, we analyzed SG morphology in late-stage WT and homozygous *hkb* mutant embryos, as well as *hkb* transheterozygous embryos over a deficiency line deleting the *hkb* locus. At stage 12, while the WT SGs turned and migrated toward the posterior of the embryo upon reaching the visceral mesoderm, SGs in *hkb* mutant embryos migrated only slightly posteriorly (*Figure 1L–N* and *Figure 1—figure supplement 4A–E*). Consistent with the role of Hkb in endoderm specification (*Brönner et al., 1994*), *hkb* mutant embryos showed defects in gut morphology, as well as defects in head development and occasional defects in germ band retraction (*Figure 1L'–N'* and *Figure 1—figure supplement 4B'-E'*), making it difficult to determine their correct stage after stage 14. Stage 15+ embryos in *hkb* mutants formed small SGs compared to stage 15 WT embryos (*Figure 1L'–N'* and *Figure 1—figure supplement 4A–E'*), as in a previous study (*Myat and Andrew, 2000b*). We observed a range of SG lengths. More than 40% of stage 15+ *hkb* null embryos and transheterozygotes of null over a deficiency (*hkb³⁵¹*, *hkb³⁹¹*, and *hkb³⁵¹/Df*) exhibited very short and ball-like SGs (*Figure 1N' and O'* and *Figure 1—figure supplement 4*). In contrast, the majority of *hkb* hypomorphic mutants and transheterozygotes of hypomorphs over a deficiency (*hkb¹²¹*, *hkb¹³¹*, and *hkb¹²¹/Df*) and other *hkb* mutants (*hkb^A321R1*, *hkb²*, and *hkb^A321R1/Df*) had intermediate SG tube lengths (*Figure 1M' and O'* and *Figure 1—figure supplement 4*), suggesting that these alleles have a residual function of *hkb*. Homozygous Hkb^mut-mCh embryos exhibited the same range of phenotypes as *hkb* null mutants, confirming that the mCh insertion disrupts Hkb function (*Figure 1—figure supplement 2*).

In all *hkb* mutant lines, one or both SGs frequently mislocalized anteriorly, toward the head center region (*Figure 1M' and N'* and *Figure 1—figure supplement 4C'-H'*). Using the salivary duct marker Dead Ringer (Dri; *Gregory et al., 1996*) and Ecad, we found that the mispositioning of the SG may be due, at least in part, to the defects of the duct, likely resulting from a combination of reduced cell number, aberrant apical domain size, and defective cell intercalation of duct cells (*Figure 1—figure supplement 5A–K'*). We also found a transient Hkb expression in the SG duct at early stage 11 (*Figure 1—figure supplement 5L–M'*), suggesting a possible role of Hkb during SG duct morphogenesis.

Together, we have generated molecularly defined null and hypomorphic alleles of *hkb* and fluorescently tagged knock-in transgenic lines. Since the two null and two hypomorphic alleles behaved similarly, we used *hkb³⁵¹* and *hkb¹²¹* for further analyses.

## Hkb is required for SG cell survival to ensure proper organ size

To determine the cellular mechanisms underlying the small SG phenotype in *hkb* mutant embryos, we measured the SG lumen length and counted SG cells using Ecad and CrebA signals, respectively. Compared to WT, *hkb* null mutants and transheterozygous *hkb* null over a deficiency showed a significantly shorter SG lumen length, and *hkb* hypomorphic embryos showed a milder effect (*Figure 2A–D and G*). Surprisingly, *hkb* null mutants also showed fewer SG cells compared to WT, with a moderate

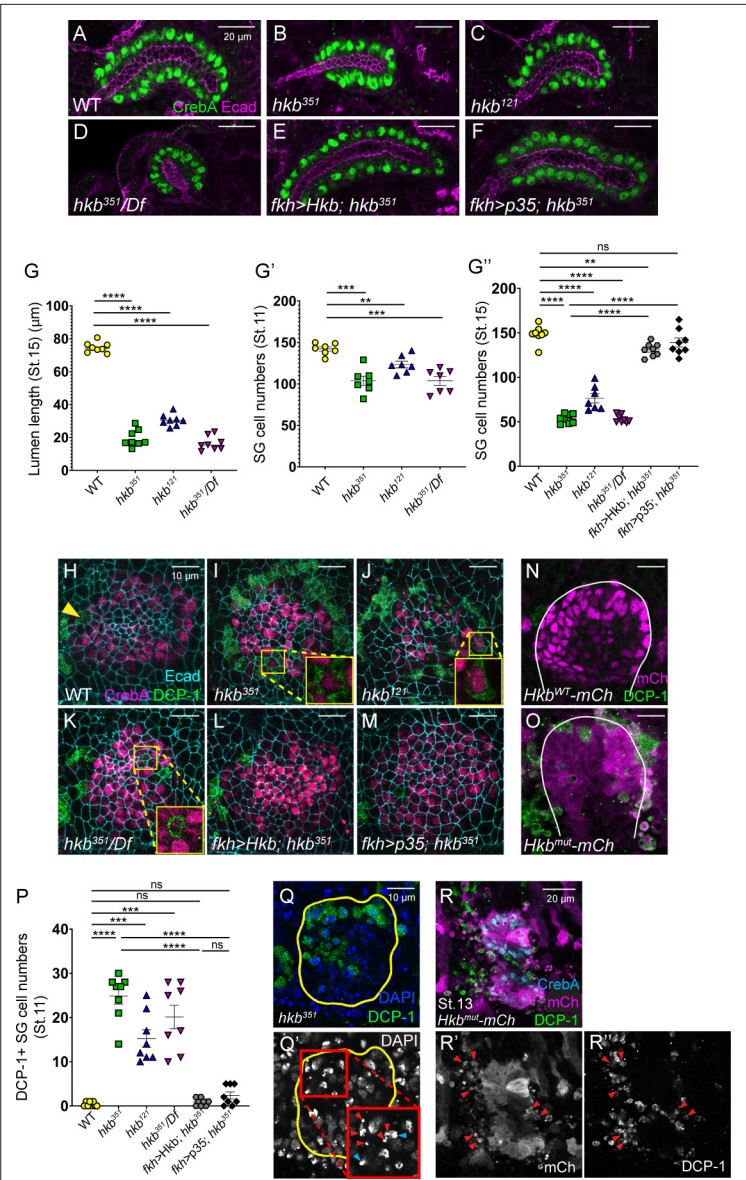

**Figure 2.** Huckebein (Hkb) is required to prevent apoptosis in salivary gland (SG) cells to ensure proper organ size. (**A–F**) Stage 15+ SGs immunolabeled for E-Cadherin (Ecad) (magenta) and CrebA (green). (**G**) Quantification of the stage 15+ SG lumen length. n=9 (WT), 12 (*hkb^351^*), 11 (*hkb^121^*), and 8 (*hkb^351^/Df*). One-way ANOVA. ****p<0.0001. (**G'-G"**) Quantification of the number of SG secretory cells at stage 11 (G'; n=7 for all genotypes) and stage 15 (G"; n=8 for all genotypes except *hkb^121^* (n=7)). One-way ANOVA. **p<0.01; ***p<0.001; ****p<0.0001. (**H–M**) Pre-invaginating SGs immunolabeled for death caspase-1 (DCP-1) (green), CrebA (magenta), and Ecad (cyan). Insets, magnified views. (**N, O**) Hkb^WT^-mCh and Hkb^mut^-mCh SGs immunostained for mCh (magenta) and DCP-1 (green). White lines, SG boundary determined by SG-specific CrebA signals. (**P**) Quantification of the DCP-1 positive SG cells. n=8 for all genotypes. One-way ANOVA. ***p<0.001; ****p<0.0001. (**Q, Q'**) Stage 11 *hkb* null mutant SG stained for DAPI (blue) and DCP-1 (green). Insets, higher magnification of the red-boxed region. Red arrowheads, fragmented nuclei. Cyan arrowheads, pyknotic nuclei. Yellow lines, SG boundary determined by SG-specific CrebA signals. (**R-R"**) SG of a stage 13 Hkb^mut^-mCh embryo immunostained for mCh (cyan), CrebA (magenta), and DCP-1 (green). Arrowheads, mCh- and DCP-1-positive cells near the SG.

The online version of this article includes the following figure supplement(s) for figure 2:

**Figure supplement 1.** Death caspase-1 (DCP-1) signals in late-stage salivary glands (SGs) and other regions of *huckebein* (*hkb*) mutant embryos.

*Figure 2 continued on next page*

*Figure 2 continued*

**Figure supplement 2.** Overexpression of Huckebein (Hkb) or p35 rescues *hkb* mutant salivary gland (SG) phenotypes.

**Figure supplement 3.** *huckebein* (*hkb*) mutant salivary glands (SGs) have a smaller apical domain size with Rab11 mislocalized to the basolateral domain.

reduction at stage 11 and a significant reduction at stage 15 (stage 15+ for *hkb* mutants) (***Figure 2G′ and G″***). The effect was milder in hypomorphic embryos (***Figure 2G′ and G″***), indicating a correlation between the SG lumen length and the number of cells.

To test whether *hkb* mutant SG cells undergo apoptosis, we labeled WT, *hkb* mutant, and Hkb^WT^-mCh and Hkb^mut^-mCh embryos with an antibody against *Drosophila* cleaved death caspase-1 (DCP-1; ***Sarkissian et al., 2014***). WT and Hkb^WT^-mCh SG cells showed no DCP-1 signals (***Figure 2H and N***), except for a single cell at the anterior boundary of the SG placode in some stage 11 embryos (Arrowhead in ***Figure 2H***). In contrast, SGs in *hkb* mutants and Hkb^mut^-mCh embryos displayed significant DCP-1 signals at stages 11 and 12 (***Figure 2I–K, O and P*** and ***Figure 2—figure supplement 1A–C′***). DAPI staining revealed condensed and fragmented nuclei in some SG cells in *hkb* mutants (***Figure 2Q and Q′***), a hallmark of pyknosis (***Zamzami and Kroemer, 1999***). At stages 13 and 14, several fragmented cells positive for DCP-1 and mCh, and negative for CrebA, were found near the SG in Hkb^mut^-mCh embryos (***Figure 2R–R″***; ***Figure 2—figure supplement 1D–E″***). Similar DCP-1-positive cells were found near the SG in *hkb* null mutants, fewer in *hkb* hypomorphs, and rarely in WT embryos (***Figure 2—figure supplement 1F–H′***). These cells, lacking CrebA signals but exhibiting nuclear fragmentation and condensation, are likely late-stage apoptotic SG cells. Dying cells were also observed in other parts of the Hkb^mut^-mCh embryos near mCh-expressing regions compared to Hkb^WT^-mCh embryos (***Figure 2—figure supplement 1I–K″***), as well as some *hkb* mutant SG duct cells (***Figure 1—figure supplement 5E, E′***), which may be the cause of the decreased duct cell numbers. This suggests a role for Hkb in cell survival in multiple cell types.

We next performed a rescue experiment. Consistent with a previous study (***Myat and Andrew, 2002***), overexpression of Hkb in SG secretory cells using the *fkh-Gal4* driver (***Henderson and Andrew, 2000***) in the otherwise WT background had no overt effects on SG formation, except for an expanded SG lumen at late stages. Overexpression of Hkb in the SG in *hkb* mutants almost completely rescued the short SG phenotype of *hkb* mutants, resulting in elongated SGs that resembled WT SGs (***Figure 2F***; ***Figure 2—figure supplement 2C-E, K***), with a reduced number of DCP-1-positive SG cells (***Figure 2L and P***), and restored lumen length and SG cell number to WT levels (***Figure 2E and G″***). Additionally, blocking apoptosis in the *hkb* mutant SG by overexpressing the baculoviral protein p35, a caspase inhibitor of apoptosis, reduced the number of DCP-1-positive cells (***Figure 2M and P***) and rescued the shortened SG phenotype, with restored SG cell number (***Figure 2G″***; ***Figure 2—figure supplement 2F-I, K***). These data suggest that the small SG size in *hkb* mutants is primarily caused by the apoptosis of SG cells.

## Hkb is required for apical domain elongation

A previous study showed that *hkb* regulates apical domain elongation by regulating the key apical protein Crb and the endocytic pathway regulator Klar (***Myat and Andrew, 2002***). Although our data showed that the reduced number of SG cells is the main reason for the small gland size in *hkb* mutants, defects in apical domain elongation may also contribute to this phenotype. Measuring the apical area of individual SG cells in stage 15 embryos revealed that *hkb* mutant SG cells have smaller apical domain sizes than WT (***Figure 2—figure supplement 3A–E***). Immunostaining with Rab11, a recycling endosome marker, revealed that while WT SGs showed Rab11 signals near the apical domain, *hkb* mutant SGs displayed basally mislocalized Rab11 signals (***Figure 2—figure supplement 3F–G′***). These data suggest that impaired apical trafficking and defective apical membrane elongation also contribute to the small gland phenotype in *hkb* mutants.

## Aberrant cell death causes centered SG invagination in *hkb* mutants

Next, we tested invagination positioning. SG-specific expression of Hkb corrected the centered invagination defect in *hkb* mutants (***Figure 3A–C***), indicating a tissue-autonomous role of Hkb in SG invagination positioning. Interestingly, blocking apoptosis in the SG also rescued the centered

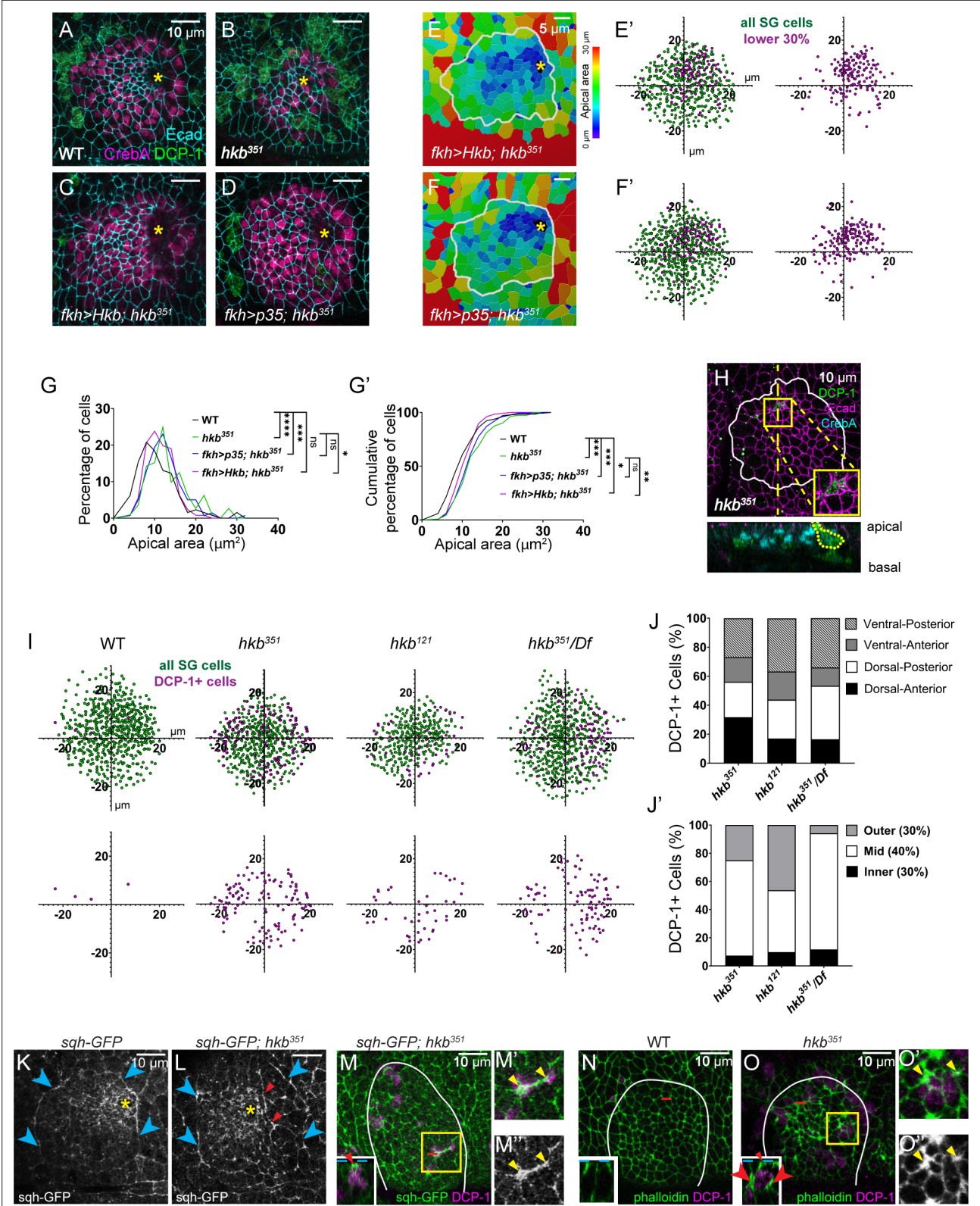

**Figure 3.** Aberrant cell death results in centered invagination in the *huckebein* (*hkb*) mutant salivary gland (SG). (**A–D**) Stage 11 SGs immunolabeled with death caspase-1 (DCP-1) (green), CrebA (magenta), and E-Cadherin (Ecad) (cyan). (**E–F**) Heat maps depicting apical area distribution in invaginating SGs. (**E'-F'**) Scatter plots showing the relative position of cells in the SG placode. X and Y axes represent the distance along the A/P and D/V axes, respectively, from the center of the placode. Cells in the bottom 30% of the apical area (magenta) and all cells (green) are plotted (n=5 SGs from five

*Figure 3 continued on next page*

*Figure 3 continued*

different embryos). (**G-G'**) Quantification of apical area distribution in invaginating SG cells. Mann-Whitney U test (percentage of cells) and Kolmogorov-Smirnov test (cumulative percentage of cells). n=5 SGs for all genotypes (WT (555 cells), *hkb*[351] (488 cells), *fkh >Hkb; hkb*[351] (526 cells)) except for *fkh >p35; hkb*[351] (6 SGs, 632 cells) *p<0.05; **p<0.01; ***p<0.001; ****p<0.0001.(**H**) *hkb* mutant SG immunostained for DCP-1 (green), Ecad (magenta), and CrebA (cyan). Insets, magnified view of the yellow boxed regions. Z-section along the dotted line displayed below. (**I**) Scatter plots showing the relative position of total SG cells (green) and DCP-1-positive cells (magenta) in stage 11 SGs. (**J,J'**) Quantification of the number of DCP-1-positive SG cells in different regions of the placode. (**K,L**) Stage 11 SGs immunostained for GFP (for *sqh-GFP*). Cyan arrowheads, supracellular myosin cable. Red arrowheads, short cables inside the SG placode. Yellow asterisk, invagination pit. (**M-M''**) Stage 11 SGs immunostained for GFP (for *sqh-GFP*; green) and DCP-1 (magenta). (**N-O''**) Stage 11 SGs stained for F-actin (green) and DCP-1 (magenta). Insets in L-N, z-sections along red dotted lines. Dotted cyan lines, apical boundary of cells. Red arrowheads/arrows indicate increased myosin (**M**) and F-actin (**O**) signals in the apical and lateral regions of the DCP-1-positive cell, respectively. (**M', M'', O', O''**) Magnified views of yellow boxed regions in **M** and **O**. Yellow arrowheads, increased myosin/F-actin signals at the apical-junctional region of DCP-1-positive cells. White lines, SG boundary determined by SG-specific CrebA signals.

invagination phenotype in *hkb* mutants (**Figure 3D**). In these embryos, cells with small apical areas were mainly distributed in the dorsal/posterior quadrant of the SG placode (**Figure 3E–F'**), as in WT (**Figure 1H and H'**). SG-specific Hkb overexpression partially rescued the apical constriction defects, and although not statistically significant, blocking apoptosis also slightly decreased the number of SG cells with small apical areas in *hkb* mutants (**Figure 3G and G'**). Since apoptotic cells undergo apical constriction before being eliminated from the epithelial sheet (**Toyama et al., 2008**; **Slattum et al., 2009**; **Marinari et al., 2012**), we hypothesized that some SG cells with small apical areas in *hkb* mutants are apoptotic. In support of this, we observed DCP-1-positive cells with reduced apical areas in *hkb* mutant SGs at stage 11 (**Figure 3H**). Among the SG cells of lower 30% of apical areas, more than 40% were DCP-1-positive (n=29/72 cells in five SGs from different embryos). We did not detect a significant spatial bias in the distribution of DCP-1-positive cells among the four quadrants of the SG placode in *hkb* null mutants, although slightly more DCP-1-positive cells were observed in the posterior region in hypomorphs and transheterozygotes of null over deficiency (**Figure 3I–J**). When we analyzed the SG placode in three domains (inner, outer, and mid), we found that most DCP-1-positive cells were in the mid domain, especially in the null and transheterozygous conditions (**Figure 3J'**). This suggests a correlation between the distribution of dying cells in the mid-domain and the widened central invagination pit observed in *hkb* mutants.

During apoptosis, the cytoskeleton actively participates in the morphological changes of dying cells. Apoptotic cells also exert a transient pulling force on the apical surface, generating the apico-basal force and increasing tissue tension (**Monier et al., 2015**). Therefore, we analyzed filamentous actin (F-actin) and myosin using phalloidin and sqh-GFP, a functional tagged version of the myosin regulatory light chain (**Royou et al., 2004**), respectively, in WT and *hkb* mutant SGs. Compared to WT, we often observed an aberrant accumulation of F-actin and myosin signals along multiple junctions within the SG placode in *hkb* mutants (**Figure 3K–O''**). In addition, there was placode-wide disruption of myosin, including the misshapen supracellular myosin cable that maintains SG tissue integrity (**Figure 3L**). Apical junctions with increased myosin or F-actin levels often coincided with DCP-1-positive cells (**Figure 3M-M'' and N–O''**). Increased F-actin expression was also seen at the apical-lateral membrane of DCP-1 positive cells (**Figure 3O**), suggesting a potential pulling force along the apicobasal axis by apoptotic cells.

## Hkb is required for the proper cell cycle status of the SG cells

To further test the impact of *hkb* loss in cytoskeletal organization, we analyzed microtubules, which play a crucial role in forming and maintaining the actomyosin network during tissue invagination (**Booth et al., 2014**; **Ko et al., 2019**; **Le and Chung, 2021**). Interestingly, immunostaining with an antibody against tyrosinated α-tubulin (Tyr-tub), which labels dynamic, short-lived microtubules (**Westermann and Weber, 2003**) revealed that, in contrast to the strong Tyr-tub signals observed at the apices of WT SG cells (**Figure 4A and A'**; **Booth et al., 2014**; **Le and Chung, 2021**), some SG cells in *hkb* mutants lacked apical Tyr-tub signals (**Figure 4B and B'**). These cells had large, rounded morphology and mislocalized nuclei near the apical surface (**Figure 4C and D**), indicating cell division. Staining with an antibody against phosphorylated histone H3 (PH3), a marker of mitotic chromatin modification (**Gurley et al., 1978**), revealed strong PH3 signals in *hkb* mutant SG cells (**Figure 4E–H and M**), indicating that these cells are actively dividing. Co-staining with Tyr-tub and PH3 revealed that mitotic

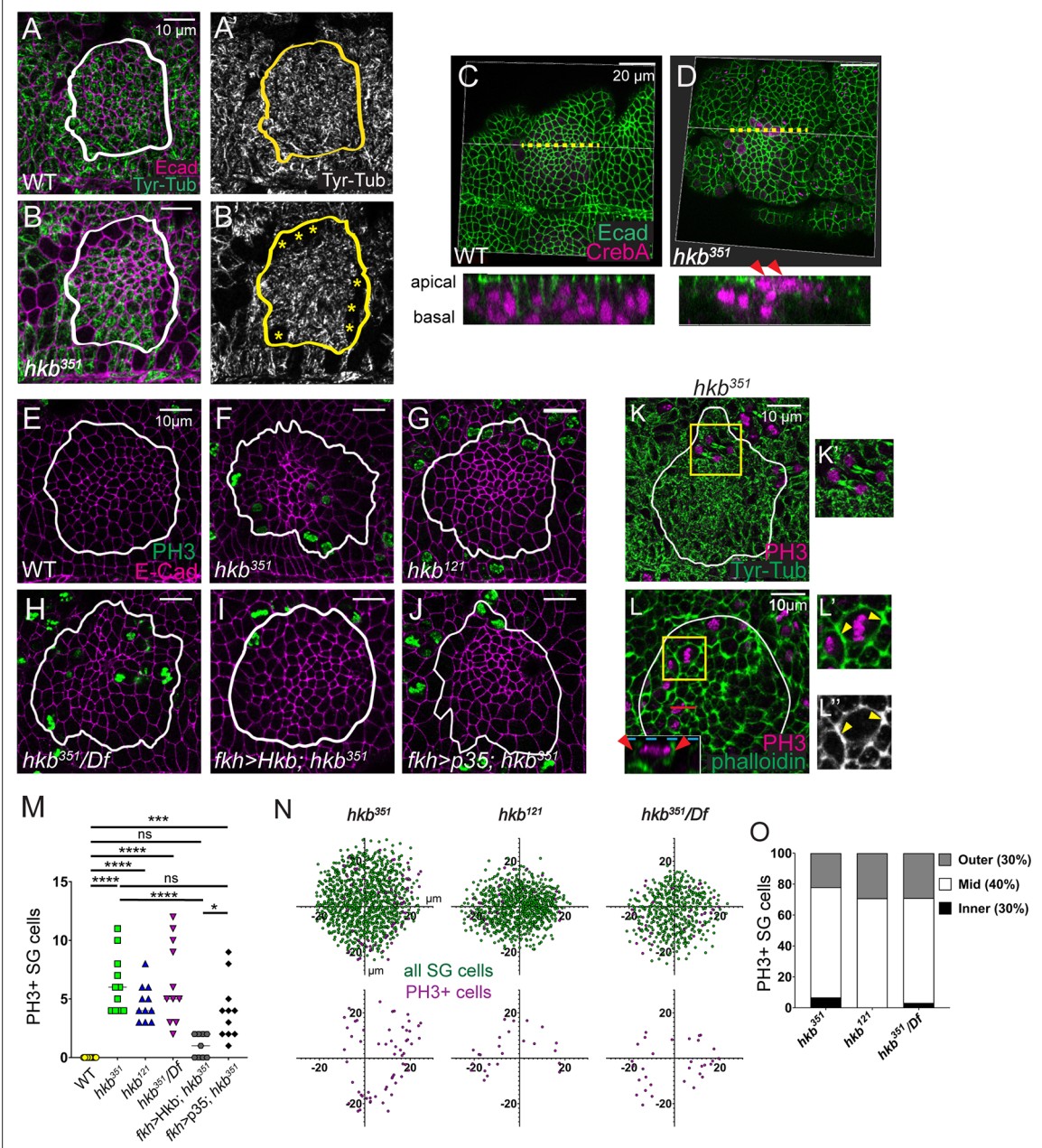

**Figure 4.** Salivary gland (SG) cells undergo abnormal cell division in *huckebein* (*hkb*) mutants. (**A-B′**) Stage 11 SGs immunolabeled with tyrosinated α-tubulin (Tyr-Tub) (green) and E-Cadherin (Ecad) (magenta). Asterisks, cells without Tyr-Tub signals. (**C, D**) 3D reconstructions of confocal images of pre-invaginating SGs immunolabeled with Ecad (green) and CrebA (magenta). Z-sections along yellow dotted lines are shown below. Arrowheads, apically localizing nuclei in *hkb* mutant SG. (**E–J**) Stage 11 SGs immunolabeled with phosphorylated histone H3 (PH3) (green) and Ecad (magenta). (**K, K′**) Stage 11 SGs immunolabeled with Tyr-Tub (green) and PH3 (magenta). Higher magnification of yellow boxed region shown in **K′**. (**L-L″**) Stage 11 SGs immunolabeled with PH3 (magenta) and phalloidin (green). Inset; lateral section (XZ; red line). Higher magnification of yellow boxed region shown in **L′** and **L″**. (**M**) Quantification of the number of PH3-positive SG cells. n=11 for all genotypes except *fkh* >Hkb; *hkb351*(n=9). One-way ANOVA. *p<0.05; ***p<0.001; ****p<0.0001. (**N**) Scatter plots showing the position of total SG cells (green) and PH3-positive cells (magenta) in the SG placode. X and Y axes represent the distance along the A/P and D/V axes, respectively, from the center of the placode (n=5 SGs from five different embryos). (**O**) Quantification of the number of PH-positive SG cells in different regions of the placode. White and yellow lines, SG boundary determined by SG-specific CrebA signals.

spindles were consistently aligned parallel to the epithelial sheet (*Figure 4K and K'*; n=35 cells in metaphase and anaphase in 16 SGs).

Mitotic cell rounding and cytokinesis employ processes that reorganize the cytoskeleton, generating forces that locally affect cell rearrangement and tissue morphogenesis (*Champion et al., 2017*; *Pinheiro and Bellaïche, 2018*). We observed an increase in F-actin along the apical junctions in PH3-positive SG cells (*Figure 4L*). Increased F-actin signals were also often detected along the apical-lateral membrane in PH3-positive cells and along the entire membrane of cells in prophase (*Figure 4L-L"*; red arrowheads in inset). While no biased distribution of dividing cells was observed across the SG placode in *hkb* mutants, dividing cells were more frequently localized in the mid-domain compared to the inner or outer domain of the SG (*Figure 4N and O*). SG-specific expression of Hkb in *hkb* mutants reduced the number of PH3-positive cells significantly (*Figure 4I and M*), suggesting a role for Hkb in inhibiting the mitotic entry of SG cells.

Given that apoptosis can induce compensatory cell proliferation in neighboring cells to maintain tissue homeostasis (*Haynie and Bryant, 1977*; *Kondo et al., 2006*; *Crucianelli et al., 2022*), we investigated whether the abnormal cell division in *hkb* mutants was a result of apoptosis. The number of PH3-positive cells in p35-overexpressing SGs in *hkb* mutants was comparable to that in *hkb* mutants alone (*Figure 4J and M*), suggesting that the abnormal cell division in *hkb* mutants may not be compensatory cell proliferation induced by apoptosis. However, we cannot rule out the possibility that compensatory cell proliferation by a few remaining dying cells still contributes to some of the abnormally dividing cells in *hkb* mutants.

## Hkb regulates endoreplication progression during SG morphogenesis

The observation of cell division in SG cells was unexpected because SG cells do not divide but rather undergo endoreplication. To further test the cell cycle status in SG cells, we used the Fly-FUCCI system, which allows the identification of different cell cycle phases based on the fluorescent signals of cytoplasmic cyclin B (CycB) and nuclear E2F transcription factor 1 (E2f1) tagged degrons (*Zielke et al., 2014*; *Figure 5A*), as well as 5-Ethynyl-2'-deoxyuridine (EdU) labeling to specifically mark S phase or endocycling cells. Prior to invagination at early stage 11, all WT SG cells exhibited strong nuclear signals for E2f1 and cytoplasmic signals for CycB (*Figure 5B–B"*) and no EdU signals (*Figure 5D*), indicating G2 phase. During late stage 11 and stage 12, distal SG cells that had completed invagination showed reduced E2f1 signals and very low levels of CycB (*Figure 5F–F"*), with high EdU signals (*Figure 5H* and *Figure 5—figure supplement 1A*), consistent with a previous finding using BrdU staining (*Chandrasekaran and Beckendorf, 2005*). By stage 13, nuclear E2f1 signals were almost completely lost in all SG cells, accompanied by low levels of CycB (*Figure 5J–J"*) and high EdU signals (*Figure 5L*). These data indicate a distal-to-proximal transition to S phase. At late stage 13, distal SG cells began to show weak E2f1 (arrowheads in *Figure 5J'*) and EdU signals (arrowheads in *Figure 5L*), and by stage 14, most SG cells had regained E2f1 signals (*Figure 5N–N"*) and lost EdU signals (*Figure 5P*) except for a few proximal cells (arrowheads in *Figure 5N' and P*), indicating a distal-to-proximal transition to G2 phase. During stages 15 and 16, almost all SG cells had accumulated E2f1 signals along with increased CycB expression (*Figure 5R–R"*) and lost EdU signals (*Figure 5T*). Interestingly, in some late-stage SGs, a few distal cells lacked E2f1 signals and showed the variable intensity of EdU signals (arrowheads in *Figure 5R–R" and T*; arrowheads in *Figure 5—figure supplement 1E*), indicating re-entry into S phase and the onset of a potential second wave of endoreplication. Taken together, our data suggest a distal-to-proximal wave of endoreplication progression during normal SG morphogenesis.

In *hkb* mutant SGs, the distal-to-proximal progression of endoreplication was disrupted. It is of note that EdU signals in *hkb* mutants were generally low in the entire embryo and absent in many tissues compared to WT, suggesting endoreplication defects in multiple tissues upon *hkb* loss (*Figure 5—figure supplement 1A–I*). Similar to WT, most cells in early stage 11 *hkb* mutant SGs exhibited strong nuclear signals for E2f1 and cytoplasmic CycB signals, with little to no EdU signal (*Figure 5C–C" and E*), except in dividing cells where CycB signals were distributed throughout the entire cell (arrowheads in *Figure 5C"*), which is consistent with the distribution of CycB during mitosis (*Jackman et al., 1995*). However, starting at late stage 11 and throughout all stages, almost all SG cells in *hkb* mutants showed nuclear E2f1 signals of varying intensity (*Figure 5G–G", K–K" and O–O"*). Consistent with this, a few randomly distributed SG cells in *hkb* mutants showed variable EdU signals, with no clear pattern as seen in WT SGs (*Figure 5E, I, M and Q*), suggesting that cells either do not enter S phase

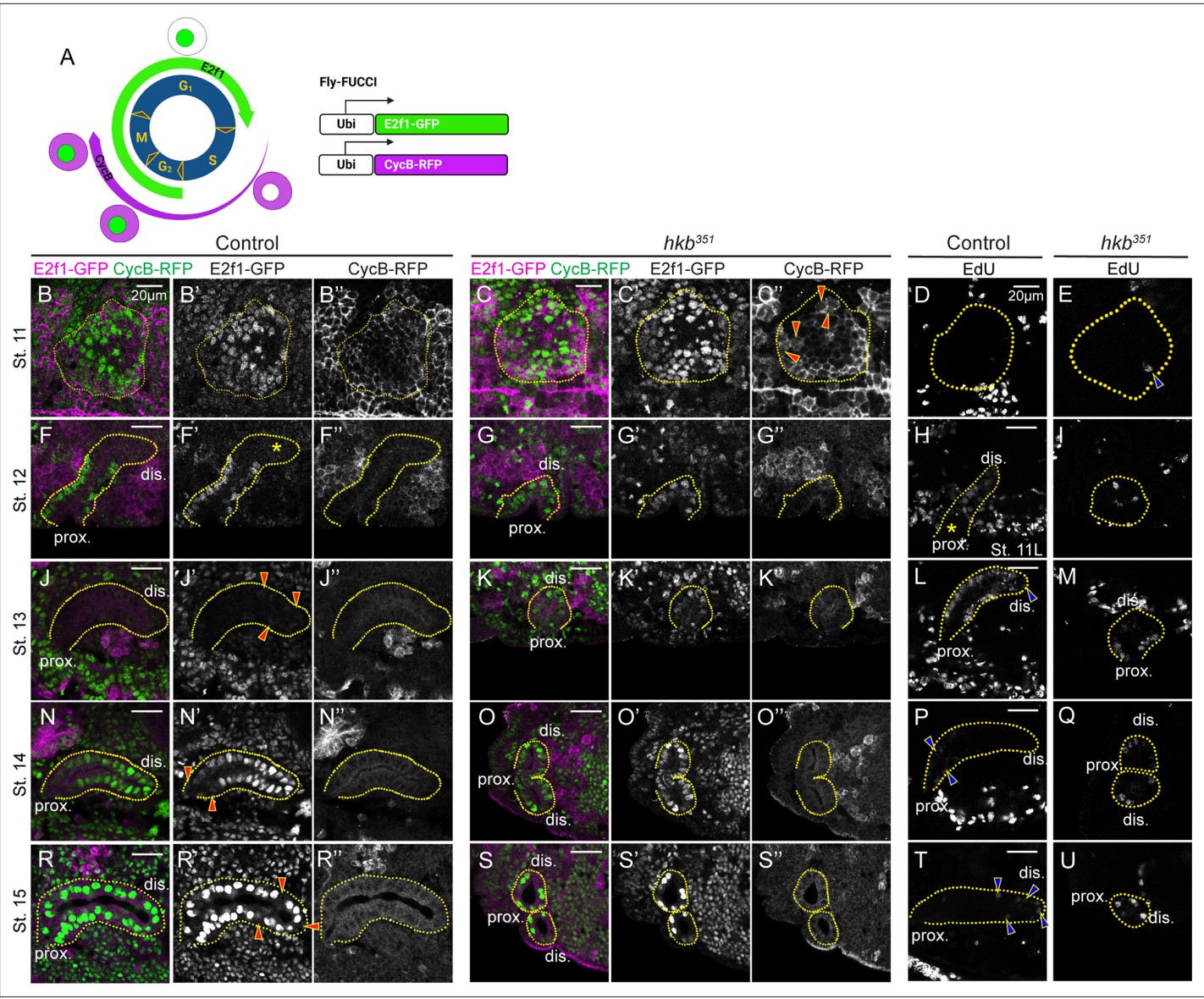

**Figure 5.** The distal-to-proximal progression of the endoreplication is disrupted in the *huckebein* (*hkb*) mutant salivary gland (SG). (**A**) Schematic of the Fly-FUCCI system. (**B-C", F-G", J-K", N-O", R-S"**) SGs immunolabeled with GFP (for E2f1-GFP; green) and cytoplasmic cyclin B (CycB) (for CycB-RFP; magenta) at different stages. Red arrowheads in **C"**, CycB signals distributed throughout the dividing cells. Asterisk in **F'**, absence of E2F transcription factor 1 (E2f1) signals in distal cells of stage 12 SG. Red arrowheads in **J'**, weak E2f1 signals in the proximal SG cells. Red arrowheads in N' and R', little to no E2f1 signal in proximal (**N'**) and distal (**R'**) SG cells. (**D–E, H–I, L–M, P–Q, T–U**) 5-Ethynyl-2'-deoxyuridine (EdU)-labeled SGs at different developmental stages. Blue arrowhead in **E**, EdU-positive single SG cell. Asterisks in **H**, absence of EdU signals in proximal cells of late stage 11 SG. Blue arrowheads in **L, P,** and **T**, weak EdU signals in distal (**L, T**) and proximal (**P**) SG cells. Yellow dotted lines, SG boundary determined by SG-specific CrebA signals. Due to the proximity of the two SGs in *hkb* mutants, one or two SGs are shown depending on the orientation of the embryo. dis. denotes distal; prox. denotes proximal.

The online version of this article includes the following figure supplement(s) for figure 5:

**Figure supplement 1.** 5-Ethynyl-2'-deoxyuridine (EdU) signals during mid to late embryogenesis in wild-type (WT) and *huckebein* (*hkb*) mutant embryos.

or enter a weak S phase. Taken together, Hkb plays a critical role in distal-to-proximal endoreplication progression during SG morphogenesis.

## Regulation of key cell cycle genes by Hkb is critical for cell survival and SG morphogenesis

To understand the mechanisms of endoreplication regulation during SG morphogenesis, we focused on the direct transcriptional targets of Hkb identified by the modENCODE Project using chromatin immunoprecipitation followed by sequencing (ChIP–seq) on the whole embryo (*Roy et al., 2010*). Gene Ontology enrichment analysis revealed several key biological processes associated with the defects observed in *hkb* mutants (*Figure 6A*). Notably, gene clusters related to the cell cycle were identified, including central players like *Cyclin A* (*CycA*), *Cyclin D* (*CycD*), *Cyclin E* (*CycE*), *E2f1*, *aurora A* (*aurA*), and *aurora B* (*aurB*). Additionally, several apoptosis-related genes were identified, including *hid*, *rpr*, and *Death related ICE-like caspase* (*Drice*). The full list of Hkb target genes, GO clusters, and the list of genes in each cluster were included in *Supplementary file 3*. Fluorescence in situ hybridization revealed a significant increase in the mRNA levels of *rpr*, *E2f1*, and *CycE* in *hkb* mutant SGs compared to WT at stages 11 and 12 (*Figure 6B–J*, and *Figure 6—figure supplement 1A–H'*), suggesting transcriptional repression of these genes by Hkb. *CycA* mRNA levels were increased in some stage 11 SG cells in *hkb* mutants, whereas all WT SG cells of the same stage showed a background level of *CycA* mRNA signals (*Figure 6—figure supplement 1I–J'*).

Although not a target of Hkb based on the Chip-seq data, we also tested to see if *hkb* acts as an indirect transcriptional inhibitor of *string* (*stg*), which encodes Cdc25 family protein phosphatase that dephosphorylates and activates Cdk1, triggering entry into mitosis in a post-blastoderm *Drosophila* embryo (*Edgar and O'Farrell, 1989*; *Gautier et al., 1991*; *Edgar et al., 1994*). Consistent with a previous study (*Edgar et al., 1994*), *stg* mRNA levels were downregulated in the WT SG once invagination began, and this remained true for the *hkb* mutant SG (*Figure 6—figure supplement 1K–M*), suggesting that the abnormal cell cycle regulation in *hkb* mutants is not due to upregulation of *stg* mRNA levels.

Disruption in cell cycle regulation can lead to apoptosis due to errors in DNA replication or failure of DNA integrity checkpoints (*Pucci et al., 2000*). Immunostaining with a marker of DNA double-strand breaks (γ-H2Av; *Lake et al., 2013*) showed increased levels in *hkb* mutant SGs compared to WT, indicating increased DNA damage (*Figure 6—figure supplement 2A–B'*). To test whether apoptosis in *hkb* mutant SG cells was triggered by the dysregulation of cell cycle genes, we analyzed double mutants for *hkb* and each of the key Hkb target cyclins and their interacting Cdk genes (*Cdk1* and *Cdk2*). Embryos single mutant for *CycA*, *CycD*, *CycE*, and *Cdk1* showed relatively normal morphology except for a few PH3-positive SG cells in *CycD* and *CycE* mutants, likely due to disrupted cell cycle and abnormal entry into mitosis (*Figure 6—figure supplement 3A–F*). Embryos double mutant for *hkb* and *CycE* or *Cdk1* rescued the *hkb* mutant phenotypes, but double mutants for *hkb* and *CycA* did not show significant rescue (*Figure 6—figure supplement 3I–M, P, Q*). *Cdk2* mutant embryos showed severe morphological defects with significantly fewer cells overall, making it difficult to assess differences in the *hkb* mutant background (*Figure 6—figure supplement 3G, H*). We confirmed the rescue effect by SG-specific knockdown experiments targeting *CycE* and *Cdk1* in *hkb* mutant SGs. As in the double mutant analysis, the knockdown of *CycE* resulted in a reduction of both dividing and dying cells (*Figure 6K, N and Q*), restored the invagination position and led to a partial rescue of the short SG phenotypes (*Figure 6K and N*; *Figure 6—figure supplement 4K–N, AB*). Similar rescue effects were observed with *Cdk1* knockdown (*Figure 6L, O, Q and R* and *Figure 6—figure supplement 4O–R, AB*).

To further test Hkb's role in cell cycle regulation, we tested Fizzy-related (Fzr), a cell cycle switch that functions with the anaphase-promoting complex (APC) to degrade mitotic cyclins during M and G1 phases and trigger exit from the M phase (*Sigrist and Lehner, 1997*). Fzr also regulates the transition from the normal cell cycle to the endocycle by degrading mitotic cyclins during G2 phase (*Sigrist and Lehner, 1997*; *Zielke et al., 2011*). *fzr* was not a ChIP-Seq target of Hkb, and the *fzr* mRNA levels in the SG were fairly high in both WT and *hkb* mutant embryos. We hypothesized that further overexpression of Fzr in *hkb* mutant SGs could suppress cell division and cell death and rescue the defective tube length phenotype. In support of our hypothesis, overexpression of Fzr in *hkb* mutant SGs resulted in a reduction of both dividing and dying cells, partially rescuing the overall *hkb* mutant

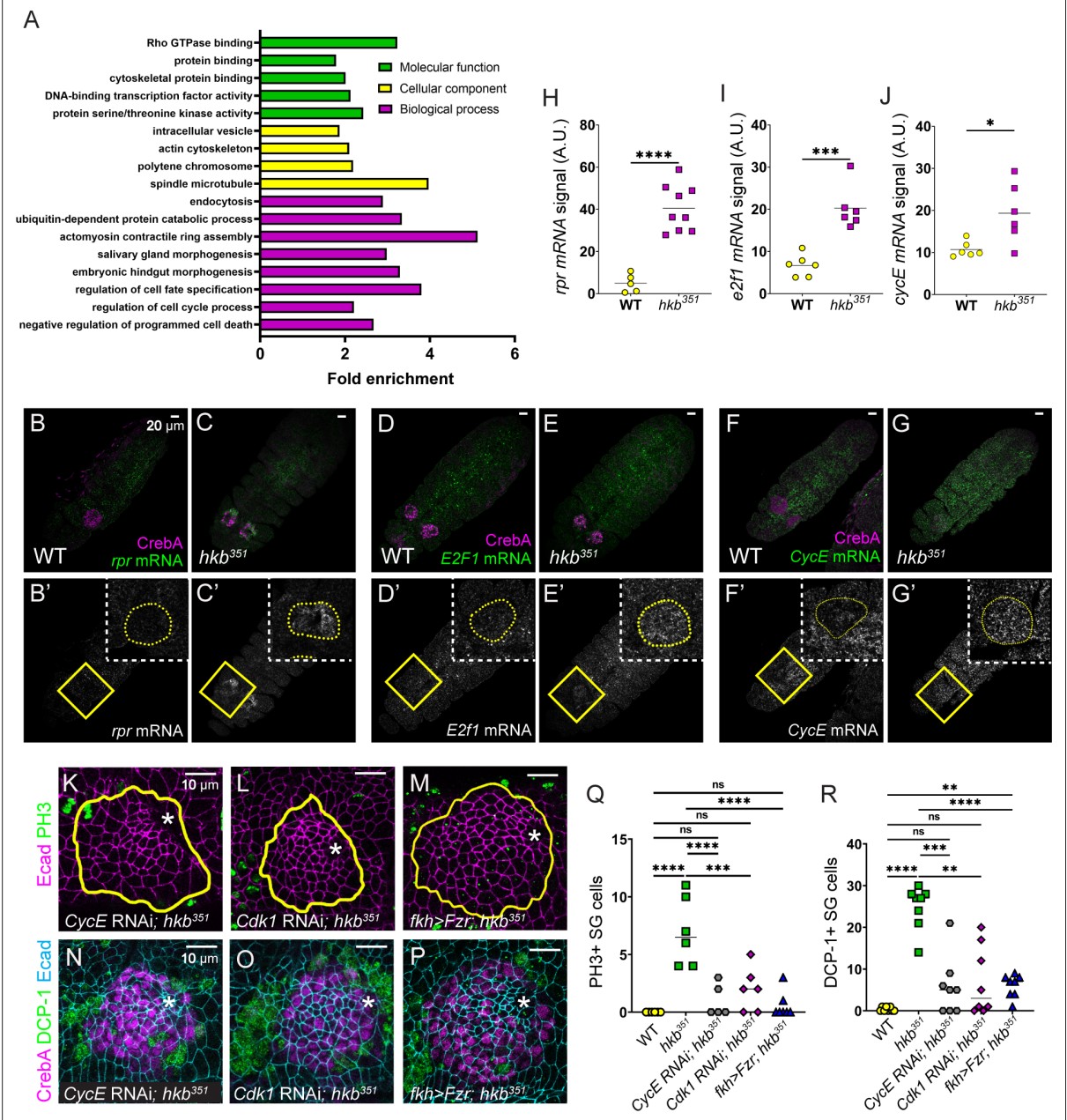

**Figure 6.** Huckebein (Hkb) regulates key cell cycle and cell death genes transcriptionally. (**A**) GO enrichment analysis of ModENCODE Hkb ChIP-Seq data. (**B-G″**) Fluorescence in situ hybridization images of stage 11 embryos showing mRNA levels of *reaper* (*rpr*) (**B-C′**), *E2F transcription factor 1* (*E2f1*) (**D-E′**), and *cyclin E* (*CycE*) (**F-G′**). Yellow-boxed regions are shown at a higher magnification in insets. (**H–J**) Quantification of the signal intensity of the mRNA levels shown in B′-G′. n=5 (wild-type, WT) and 9 salivary glands (SGs) (*hkb³⁵¹*) (**H**); 6 SGs for both genotypes (**I, J**). Student's t-test with Welch's correction. *p<0.05; ***p<0.001; ****p<0.0001. (**K–P**) *Cdk1* and *CycE* knockdowns and Fzr overexpression in *hkb* mutant SGs immunostained for E-Cadherin (Ecad) (magenta) and phosphorylated histone H3 (PH3) (green) (**K–M**) and Ecad (magenta), death caspase-1 (DCP-1) (green), and CrebA (cyan) (**N–P**). Asterisks, invagination pit. (**Q, R**) Quantification of the number of PH3-positive (**Q**) and DCP-1-positive (**R**) SG cells. n=6 (**Q**) and 8 (**R**) SGs from different embryos for all genotypes. One-way ANOVA. **p<0.01; ***p<0.001; ****p<0.0001. Yellow lines and yellow dotted lines, SG boundary determined by SG-specific CrebA signals.

The online version of this article includes the following figure supplement(s) for figure 6:

**Figure supplement 1.** Expression levels of key cell cycle and cell death genes in wild-type (WT) and *huckebein* (*hkb*) mutant embryos.

**Figure supplement 2.** γ-H2Av signals are increased in *huckebein* (*hkb*) mutant salivary glands (SGs) at stage 11.

**Figure supplement 3.** Genetic inhibition of the function of key cell cycle genes is sufficient to suppress cell death and cell division in *huckebein* (*hkb*) mutants.

**Figure supplement 4.** Inhibition of cell division or cell death partially rescues the salivary gland (SG) phenotypes of *huckebein* (*hkb*) mutants.

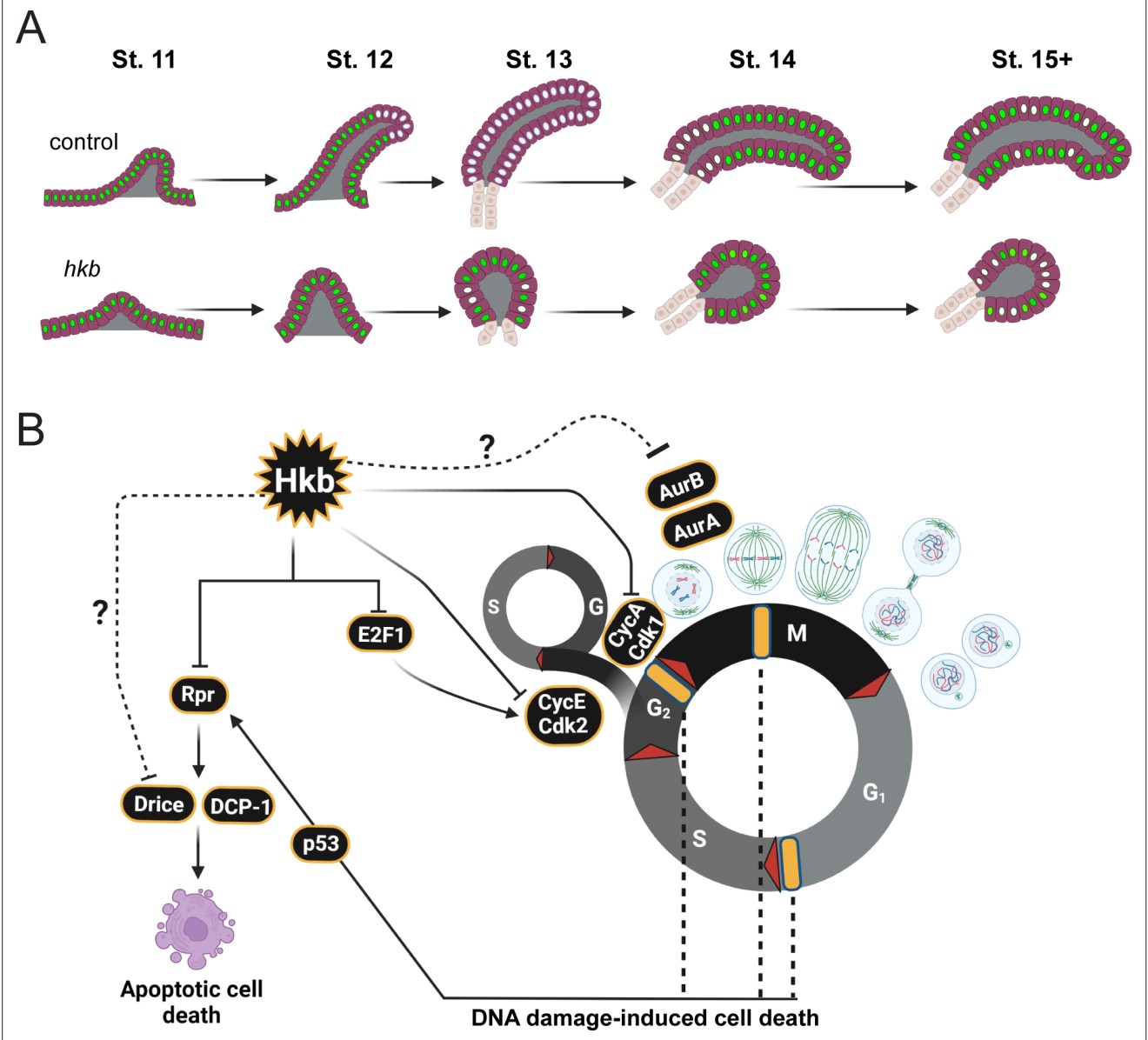

**Figure 7.** A model of endoreplication regulation by Huckebein (Hkb) during salivary gland (SG) morphogenesis. (**A**) Cartoon diagram showing the cell cycle status and progression of endoreplication during SG morphogenesis. Cells with green nuclei represent cells with E2F transcription factor (E2f1) signals (G phase). Cells with white nuclei represent cells with 5-Ethynyl-2'-deoxyuridine (EdU signals S phase). (**B**) Summary of the role of Hkb in cell cycle regulation in the SG. Hkb helps to inhibit cell division and cell death and promotes endoreplication progression through transcriptional inhibition of key cell cycle and pro-apoptotic genes. Cell death in *hkb* mutant SG may also be caused by DNA damage-induced cell death. Yellow bars represent cell cycle checkpoints.

phenotypes (*Figure 6M, P, Q and R* and *Figure 6—figure supplement 4S-V, AB*). When both Fzr and p35 were overexpressed in the *hkb* mutant background to inhibit both mitotic cyclins and cell death, we observed an enhanced rescue effect (*Figure 6—figure supplement 4W-Z, AB*). Overall, these data suggest that the loss of *hkb* leads to the derepression of cyclin/Cdk activities, which disrupts endoreplication and causes SG cells to enter mitosis abnormally (*Figure 7*). Consequently, cell death occurs in SG cells, leading to disrupted SG morphology.

## Discussion

In this study, we have uncovered a previously uncharacterized role for Hkb in coordinating cell cycle regulation and morphogenesis during development. Through a comprehensive analysis of mutant phenotypes, modENCODE ChIP-Seq targets of Hkb, and genetic interaction analyses, we have revealed the role of Hkb in regulating endoreplication to enable the proper development of the SG.

### Hkb regulates the cell cycle through transcriptional and possibly post-translational mechanisms

During *Drosophila* embryonic development, many tissues arrest the cell cycle in the G2 phase after the initial cell divisions (*Hartenstein and Campos-ortega, 1985*; *Edgar and O'Farrell, 1989*). Subsequently, several tissues, including the SG, switch to the endocycle. This switch is correlated with the development of three-dimensional organs, suggesting a metabolic need to inhibit active cell proliferation to allow for such a transition during development. Using the Fly-FUCCI system and EdU labeling, we observed progressive endoreplication from distal to proximal regions during normal SG morphogenesis (*Figure 5*). Our data are consistent with a previous study showing that distal cells enter S phase first during SG invagination (*Chandrasekaran and Beckendorf, 2005*). Notably, our observation that some distal SG cells in WT re-enter S phase during late SG morphogenesis suggests a potential second wave of endoreplication toward the end of embryogenesis that has not been described. In *hkb* mutant SGs, progressive endoreplication is lost.

We propose that endoreplication in SG cells is regulated, at least in part, by Hkb-mediated transcriptional repression of *E2f1* and *CycE* (*Figure 6* and *Figure 6—figure supplement 1*). E2f1 is usually destroyed as cells transit into S phase through degradation by the S-phase activated E3 ubiquitin ligase Cullin 4. Whenever E2f1 expression persists in S phase, as is seen with ectopic expression or overexpression of E2f1, cell cycle arrest is disrupted and the rate of cell cycle progression is increased, leading to a significantly higher number of cells in the mitotic phase (*Neufeld et al., 1998*; *Shibutani et al., 2008*), as we observed in the *hkb* mutant SG. The changes in the cell cycle control mechanism caused by persistent E2f1 expression during the S phase ultimately lead to widespread apoptosis in affected tissues (*Shibutani et al., 2008*). During normal SG development, E2f1 and CycE (which is also a transcriptional target of E2f1; *Ohtani et al., 1995*; *Duronio and O'Farrell, 1995*; *Royzman et al., 1997*; *Duronio et al., 1998*) may together control the transition to S phase to begin endoreplication. We propose that Hkb promotes SG cell progression through the S phase, at least in part, through its repression of *E2f1* and *CycE* expression. In support of this, knockdown of *CycE* partially rescued the *hkb* mutant phenotypes (*Figure 6*).

Notably, *stg* transcript levels remain low in *hkb* mutant SGs, suggesting that the abnormal mitotic entry of *hkb* mutant SG cells is not due to *stg* upregulation. We propose that Hkb prevents entry into mitosis, at least in part, by downregulating *CycA* expression. CycA acts in the *Drosophila* G2/M phases of the cell cycle, where it regulates aspects of the DNA replication mechanism and entry into mitosis by binding to Cdk1 (*Pagano et al., 1992*; *Knoblich et al., 1994*; *Yam et al., 2002*; *Gong and Ferrell, 2010*; *Bendris et al., 2011*; *Sallé et al., 2012*). Loss of *hkb* may prevent inhibition of CycA-Cdk1 activity in the SG. In support of this idea, the knockdown of *Cdk1*, or overexpression of *Fzr,* an inhibitor of mitotic cyclins, partially rescued the *hkb* mutant phenotypes (*Figure 6*). Depletion of the zygotic *CycA* pool did not rescue the *hkb* mutant phenotypes, either because of the contribution of the maternal *CycA* pool or because of the presence of CycB and CycB3, two other cyclins that bind to Cdk1 to regulate mitotic entry.

Hkb may regulate the cell cycle not only through transcriptional control of cell cycle genes but also by influencing ubiquitination, a key mechanism of cell cycle regulation. ChIP-Seq analysis revealed that Hkb targets several key components involved in ubiquitination, including *cullin 1* and *archipelago* (*ago*) (*Figure 6A*; *Supplementary file 3*). Ago is known to regulate CycE levels in *Drosophila* (*Moberg et al., 2001*). Remarkably, ZnF transcription factors can interact with both nucleic acids and proteins and are important regulators of protein degradation through ubiquitination (*Gibson et al., 1988*; *Iuchi and Kuldell, 2005*). Further studies will provide a deeper understanding of the comprehensive mechanisms by which Hkb regulates the cell cycle.

## Progressive endoreplication during organ formation

Endoreplication in the *Drosophila* embryonic SG, coupled with invagination, allows cells to increase the number of gene copies to provide sufficient proteins for morphogenesis. The regulatory mechanism behind the progressive distal-to-proximal endoreplication in the SG remains unclear. In agreement with *Sánchez-Corrales et al., 2021*, we observed higher levels of Hkb in the posterior region of the placode at stage 10, which became more uniformly distributed throughout all SG cells by stage 11 (*Figure 1—figure supplement 5*). The initial difference in Hkb levels within the SG placode may lead to the differential repression of cell cycle genes, contributing to the progression of endoreplication. Alternatively, SG invagination in the dorsal/posterior region could potentially trigger the sequential onset of endoreplication, as suggested by a previous study (*Chandrasekaran and Beckendorf, 2005*). We observed a disruption in the progressive entry of the SG cells into endoreplication when Hkb function is lost. The degree of disruption varied between samples, with some embryos showing no SG cells in S phase, while others had a small number of SG cells at random positions along the proximal-distal axis with weaker EdU signals compared to WT SGs of similar stages, suggesting that they may be entering a weak S phase (*Figure 5*). Further investigation of the relationship between endoreplication regulation and SG invagination will provide a better understanding of how cell cycle and morphogenesis coordinate during organ formation. It will also be interesting to test a role for Hkb in cell cycle (endocycle) regulation in other *Drosophila* tissues, and to test a similarly conserved role for the mammalian SP/KLF transcription factors.

## Hkb's roles in cytoskeletal reorganization and intracellular trafficking

In line with previous studies on the cellular forces exerted by dividing (*Kondo and Hayashi, 2013*; *Pinheiro and Bellaïche, 2018*) and apoptotic (*Monier et al., 2015*; *Toyama et al., 2008*) cells on neighboring cells, the abnormal cell division, and excessive cell death observed in *hkb* mutant SGs are associated with a tissue-wide disruption of actomyosin and cell geometry (*Figure 3*). Our data show that the centralized invagination phenotype results primarily from excessive cell death (*Figure 2* and *Figure 3*). However, dividing cells, particularly those near the SG boundary, may contribute to this phenotype by undergoing cell surface area expansion during cell division, increasing the number of cells with larger apical areas along the SG boundary (*Figure 4*).

Based on Hkb ChIP-Seq targets, we propose that Hkb also plays an active role in regulating cytoskeletal organization through transcriptional regulation of key cytoskeletal regulators. Hkb targets include key Rho signaling components and their regulators, such as the small GTPases Rho1 and Rac1, and several RhoGEFs, including RhoGEF2, a key activator of Rho signaling during epithelial morphogenesis (*Barrett et al., 1997*; *Figure 6A*; *Supplementary file 3*). We also propose a critical role for Hkb in regulating intracellular trafficking. Supporting previous research on Hkb's role in regulating Klar, an important regulator of intracellular trafficking (*Myat and Andrew, 2002*), our data show that Rab11 is mislocalized basally in *hkb* mutant SGs (*Figure 2—figure supplement 2*). Multiple endocytic components, including *βCOP*, *AP-1-2β*, *AP-2α*, and *Nuf*, as well as several Rab GTPases, including *Rab4*, *Rab10*, *Rab27*, and *Rab30*, were also identified as Hkb ChIP-Seq targets (*Figure 6A*; *Supplementary file 3*), providing further evidence for the involvement of Hkb in intracellular trafficking. The initial difference in Hkb levels in the SG placode may lead to differential expression of these genes, which could contribute to normal invagination positioning.

In conclusion, we have uncovered a role for the Hkb transcription factor in regulating endoreplication and preventing abnormal cell cycle progression during epithelial tube formation. Our findings provide valuable insights into a critical mechanism for coordinating cell cycle regulation and morphogenesis during organ formation.

# Materials and methods
## Fly stocks and genetics

The fly stocks used in this study are listed in *Supplementary file 1*. All the crosses were performed at 25 °C, except for Hkb expression using fkh-Gal4 and UAS-Hkb for the rescue experiment, which was performed at 29 °C for higher levels of transgene expression.

## Generation of *hkb* mutant alleles

*hkb* mutant alleles were generated using CRISPR/Cas9. Using flyCRISPR Target Finder (http://target-finder.flycrispr.neuro.brown.edu/), we identified two gRNAs, one in the first exon of the *hkb* transcript near the 5' end of the gene (gRNA3) and the other in the second zinc finger domain (gRNA1). Both gRNAs exhibited maximum stringency, with zero predicted off-target effects. The sense and antisense oligos for each gRNA sequence were annealed and cloned into the pU6-BsaI-gRNA vector (a kind gift from Jean-Paul Vincent; *Baena-Lopez et al., 2013*) via the BsaI restriction site. The constructs were confirmed by Sanger sequencing before injection. The following sense and antisense oligonucleotides were used for creating null (gRNA3) and hypomorphic (gRNA1) alleles (PAM sequences underlined and linkers in small characters).

> gRNA1 sequence: 5'-ACGCGTCAGTTCCTCATTGC<u>GGG</u>-3'
> sense oligo, 5'-tccACGCGTCAGTTCCTCATTGC-3'
> antisense oligo, 5'-aaacGCAATGAGGAACTGACGCGT-3'
> gRNA3 sequence: 5'-AAAGTCGCGAGTAGGTTTGC<u>GGG</u>-3'
> sense oligo, 5'-tccAAAGTCGCGAGTAGGTTTGC-3'
> antisense oligo, 5'-aaacGCAAACCTACTCGCGACTTT-3'

The pU6-gRNA1/3 constructs (500 ng/µl per gRNA) were sent to GenetiVision for injections. GenetiVision performed injection into nos-Cas9 embryos. Individual G0 adults were crossed to w; TM3, Sb/TM6B, Tb flies. Five offspring (F1) for each cross were independently crossed to w; TM3, Sb/TM6B, Tb to generate independent stocks. The DNA fragment encompassing the gRNA sites was PCR amplified using a single adult fly of each stock, and indels at the corresponding gRNA site were confirmed by Sanger sequencing. The following primers were used for both PCR and sequencing.

> Hkb_FWD, 5'-AGATTGGGTTTGGTGAGTGC-3'
> Hkb_REV, 5'-TCGCACATCCTCAGACAGAC-3'

The predicted protein sequences and details of indels for each allele were described in *Figure 1—figure supplement 1*.

## Generation of Hkb-mCherry fusion protein lines

To create the functional Hkb-mCh fusion protein line, mCherry was inserted into six amino acids upstream from the C-terminal end of the Hkb protein in frame with the Hkb protein using the homology-directed repair of the CRISPR/Cas9 technique. The gRNA near the C-terminal end of the protein was chosen (Hkb_C_gRNA3, 5'-CCCATGTACTCATACCTTTATGG-3') using the flyCRISPR Target Finder (http://targetfinder.flycrispr.neuro.brown.edu/). Genomic DNA regions that flank the gRNA locus were amplified using the WT genomic DNA for the left and right homology arms (HAL (1066 bp) and HAR (1408 bp)). DNA fragments for the open reading frame of the mCherry protein and the backbone plasmid were amplified using the pENTR R4-mCherry-R3 plasmid (Addgene #32311) as a template. The donor DNA construct containing both homology arms (pENTR-HAL-mCherry-HAR; 5853 bp) was created using the NEBuilder HiFi DNA Assembly Master Mix (NEB #E2621). The following primers were used for amplifying each DNA fragment.

> mCh_FWD, 5'-CCATCTTCGTGCCCATGTACATGGTGAGCAAGGGCGAG-3'
> mCh_REV, 5'-CTTGTACAGCTCGTCCATGC-3'
> Backbone_FWD, 5'-CTATGCCGTCTCCATTCGATACAACTTTGTATAATAAAGTTGAAC-3'
> Backbone_REV, 5'-GTTGGCCAACTTTGTATAG-3'
> HAL_FWD, 5'-TCTATACAAAGTTGGCCAACAGATCTCGCAGCACACAC-3'
> HAL_REV, 5'-GTACATGGGCACGAAGATG-3'
> HAR_FWD, 5'-GCATGGACGAGCTGTACAAGTCATACCTTTATGGCTACTGAG-3'
> HAR_REV, 5'-ATCGAATGGAGACGGCATAG-3'

The same strategy was used to create the Hkb^mut^-mCh knock-in line. The mCh tag was inserted into the N-terminal region of Hkb, using gRNA3, the same gRNA used to create null mutant alleles. A stop codon was added to the C-terminal end of the mCh tag. Genomic DNA regions that flank the gRNA locus were amplified using the WT genomic DNA for the left and right homology arms (HAL (1130 bp) and HAR (1153 bp)). The following primers were used for amplifying each DNA fragment.

mCh_FWD, 5'- CCCGCAAATGGTGAGCAAGGGC –3'
mCh_REV, 5'- GCGAGTAGGTCTACTTGTACAGCT-3'
Backbone_FWD, 5'- AAATATTGAGATACAACTTTGTATAATAAAGTTGAACGAGAAACGTAAAA –3'
Backbone_REV, 5'- TGGCTGCAGTTGGCCAACTTTGTATAGAAAAGTTGAAC –3'
HAL_FWD, 5'- GGCCAACTGCAGCCAGAATCGCG -3'
HAL_REV, 5'- CTCACCATTTGCGGGGGATG -3'
HAR_FWD, 5'- TACAAGTAGACCTACTCGCGACTTTTCC-3'
HAR_REV, 5'- TATACAAAGTTGTATCTCAATATTTGCATATATCGTCGAGGTCCAAAGA -3'

The pENTR-HAL-mCherry-HAR constructs (500 ng/µl per gRNA) were sent to GenetiVision for injections. GenetiVision performed injection into nos-Cas9 embryos. Individual G0 adults were crossed to w; TM3, Sb/TM6B, Tb flies. Five offspring (F1) from each of the surviving crosses were independently crossed to w; TM3, Sb/TM6B, Tb to generate independent stocks. Fluorescent signals for mCherry in Hkb$^{WT}$-mCh and Hkb$^{mut}$-mCh transgenic lines were confirmed by immunostaining using the antibody against mCherry.

## Immunostaining and imaging

*Drosophila* embryos were collected on grape juice agar plates at 25 °C. Embryos were dechorionated by incubating in 50% bleach for 3 min. For most samples, embryos were fixed in formaldehyde-saturated heptane for 40 min and devitellinized using 80% ethanol. Following this, the embryos were stained with primary and secondary antibodies in PBSTB (1 X PBS, 0.2% bovine serum albumin, and 0.1% Triton X-100). For phalloidin and sqh-GFP signals, embryos were hand-devitellinized and stained with primary and secondary antibodies in PBTr (1 X PBS and 0.1% Triton X-100). The antibodies used in this study and their respective dilutions are listed in *Supplementary file 2*. After staining, embryos were mounted in Aqua-Poly/Mount (Polysciences, Inc) and imaged with Leica SP8 confocal microscope using 63 x, NA 1.4, and 40 x, NA 1.3 objectives. For horseradish peroxidase (HRP) staining, the VECTASTAIN ABC-HRP Kit, Peroxidase (PK-4000; Vector Laboratories) was used with biotin-conjugated secondary antibodies. Whole embryo HRP images were acquired with a Leica DM2500 microscope using a 20x, 0.8 NA objective.

## Cell segmentation, apical area quantification, and scatter plots

SGs in early (pre-invagination; invagination depth, 0 µm) and late (during invagination; invagination depth, 5–10 µm) stage 11 of embryos were used. Confocal images of the SG were obtained from embryos immunostained for the Ecad (adherens junctions) and CrebA (SG cell nuclei) and used for cell segmentation. A maximum intensity projection was generated using the LasX program from three apical z-sections of confocal images of SG cells encompassing the adherens junction. Using the maximum intensity projection, cells were segmented using Ecad signals as a marker for each cell boundary using Imaris software (Andor). The apical area was calculated using Imaris and individual cells were color-coded based on the size of their apical areas. The frequency distribution of the apical area of the cells was analyzed using GraphPad Prism with the data exported from Imaris. Since the output data from Imaris for the apical area quantification is a sum of the area of both sides of the cell layer, we further divided the values by 2 to obtain the apical area of SG cells.

Scatter plots were generated using the X and Y coordinates for each SG cell, which were determined with respect to the A/P and D/V axes of the tissue using Imaris. The cell at the center of the SG placode was used as the origin point (X=0, Y=0).

## Quantification of SG cell numbers

Confocal microscopy stacks for SGs with SG-specific nuclear markers (CrebA or Sage) were used for early stage 11 (pre-invagination) and stage 15+ embryos. 3D-reconstructed images were generated using Imaris. The number of nuclei for each 3D-reconstructed SG was counted using cell ID numbers.

## Quantification of SG lumen length and apical area of individual SG cells at stage 15

SG lumen length at stage 15 was measured for the secretory portion of the gland based on CrebA signals, which are exclusively expressed in the secretory cells at this stage. Confocal images immuno-labeled with markers for Ecad and CrebA for stage 15 SGs were used. 8–12 z-section images of the SG were merged to obtain a maximum projection image clearly showing half of the SG lumen and its apical surface. Segmentation of SG cells and quantification of apical areas of individual cells were performed using Imaris as described for segmentation of SG cells at stage 11.

For the lumen length quantification, maximum intensity projection images were imported into the ImageJ software (Fiji). The length of the SG lumen was measured along the line following the midpoint of the secretory portion of the tube.

## Quantification of SG phenotypes at embryonic stages 12 and 15+

After performing HRP staining with CrebA, the SGs were manually categorized based on their pheno-types. At stage 12, WT SGs turn and migrate towards the posterior of the embryo after reaching the visceral mesoderm. SGs showing such normal behavior were categorized as 'normal,' while those SGs that reached the visceral mesoderm and migrated slightly were categorized as 'turned but short,' and those SGs that reached the visceral mesoderm but did not migrate posteriorly were categorized as 'not turned'.

For stage 15+ embryos, SGs that were comparable in shape, size, and position to WT SGs were categorized as 'normal;' SGs that had a rounded/ball-like morphology were categorized as 'very short/rounded;' SGs that were slightly elongated but not as long as WT SGs and mislocalized ante-riorly were categorized as 'intermediate, mislocalized;' SGs that were slightly elongated but in the normal position were categorized as 'intermediate, normal position.' SGs that were as long as WT but mislocalized anteriorly were categorized as 'long, mislocalized'.

## Analysis of modENCODE Hkb ChIP-Seq targets

The dataset was downloaded from the ENCODE Portal (https://www.encodeproject.org/). Total 1268 ChIP-Seq targets were analyzed using Database for Annotation, Visualization, and Integrated Discovery (DAVID; *Dennis et al., 2003*).

## EdU labeling

To visualize SG cells in the S phase of the cell cycle or undergoing endoreplication, dechorionated embryos were incubated in n-octane (Thermo Scientific Chemicals: 325950010) for 3 min to permeabi-lize the vitelline membrane. Embryos were then washed five times in 1 x PBS in a microcentrifuge tube before being pulsed with 100 μM EdU (Thermo Fisher: C10637) for 30 min. Following EdU labeling, embryos were washed five times for 5 min each in PBSTB and incubated with azide-conjugated Alexa Fluor 488 for 30 min at room temperature. Embryos were further stained with CrebA and Crumbs antibodies to mark SG cell nuclei and apical cell boundaries, respectively.

## Fluorescence in situ hybridization

Dechorionated *Drosophila* embryos were fixed in paraformaldehyde-heptane solution for 30 min, followed by devitellinization using methanol. Freshly fixed embryos were then labeled with digoxi-genin (DIG)-labeled antisense probes following the standard procedure described in *Knirr et al., 1999*. After rehydration and post-fixation in 10% formaldehyde, embryos were pre-hybridized for 1 hr at 55 °C. mRNA probes were added to the embryos which were then incubated for 16 hr at 55 °C. DIG present on the synthesized mRNA probes was subsequently labeled with anti-DIG primary antibody (Enzo; ENZ-ABS302-0001; 1:1,000) and biotin-conjugated secondary antibody (Invitrogen; 1:500). To detect fluorescent signals, embryos were incubated with AB solution (PK-6100, Vectastain Elite ABC kit) followed by the tyramide signal amplification reaction (Cy3 fluorescent dye in amplification diluent; 1:50) (Perkin Elmer Life Sciences, NEL753001kt). To mark the SG and the apical membrane, the embryos were immunolabeled with antibodies for CrebA (SG nuclei) and Crb (apical membrane).

Primer sequences used for mRNA probe synthesis in this study are listed below.

| Gene name | Primer sequence | Probe size |
|---|---|---|
| Cyclin A (CycA) | Forward: 5'-TAAACCACGAACCGCTGAAC-3'<br>Reverse: 5'-CATGGTGCGCTCTTTCCTAC-3' | 443 bp |
| Cyclin E (CycE) | Forward: 5'-TGGACTGGTTGATCGAGGTC-3'<br>Reverse: 5'-TCAAAGCCAGAGAATTGCGG-3' | 424 bp |
| E2f1 | Forward: 5'-CAGTCCGCAAAAATGTGAAA-3'<br>Reverse: 5'-TATCACACGCAACCACAACA-3' | 500 bp |
| fizzy-related (fzr) | Forward: 5'-ACCCCACAGCAGCAATAGAT-3'<br>Reverse: 5'-TTCGGGGGTATGAGTGTTTC-3' | 417 bp |
| reaper (rpr) | Forward: 5'-CGAAAGAAAAGTGTGTGCGC-3'<br>Reverse: 5'-GTTGTGGCTCTGTGTCCTTG-3' | 483 bp |
| string (stg) | Forward: 5'-GCGACTGATGCTGTAGGTTG-3'<br>Reverse: 5'-AGGGGTCTGATTTCGGGATC-3' | 407 bp |

## Quantification of in situ hybridization signals

Stage 11 *hkb* mutant and WT SGs were used. A maximum intensity projection was generated using two z-sections of confocal images of SG cells. Z-sections were obtained from the same depth in all instances. Snapshots of these sections were imported into Fiji (NIH), and SG *mRNA* expression levels were measured by obtaining the mean gray value of the fluorescence intensity across the SG placode. SG placode-specific measurements were performed using SG-expressed CrebA signals. For each *mRNA* quantification, background subtraction was done using 10–15 cells in the epidermis in the lateral ectoderm region of parasegment 2 near the SG. Statistical analysis was performed using GraphPad Prism.

## Acknowledgements

We thank the members of the Chung laboratory for their comments and suggestions. We thank the Bloomington Drosophila Stock Center for fly stocks, and DJ Andrew and the Developmental Studies Hybridoma Bank for antibodies. We thank Flybase and the modENCODE transcription consortium for the gene information and the Hkb ChIP-Seq data. We are grateful to DJ Andrew, J Kim, JC Larkin, and JL Woodward for their helpful comments on the manuscript. Cartoons in *Figures 1, 5 and 7* were created using BioRender. This work was supported by a start-up fund from Louisiana State University and grants from the Louisiana Board of Regents Research Competitiveness Subprogram (LEQSF(2019-22)-RD-A-04) and the National Science Foundation (MCB 2141387) to SC.

## Additional information

### Funding

| Funder | Grant reference number | Author |
|---|---|---|
| Louisiana State University | | SeYeon Chung |
| Louisiana Board of Regents | LEQSF(2019-22)-RD-A-04 | SeYeon Chung |
| National Science Foundation | MCB 2141387 | SeYeon Chung |

The funders had no role in study design, data collection and interpretation, or the decision to submit the work for publication.

### Author contributions

Jeffrey Matthew, Conceptualization, Formal analysis, Investigation, Writing - original draft, Writing - review and editing; Vishakha Vishwakarma, Thao Phuong Le, Formal analysis, Investigation, Writing - original draft; Ryan A Agsunod, Investigation; SeYeon Chung, Conceptualization, Formal analysis, Supervision, Funding acquisition, Investigation, Writing - original draft, Writing - review and editing

**Author ORCIDs**
Jeffrey Matthew (iD) http://orcid.org/0000-0001-5686-6370
SeYeon Chung (iD) http://orcid.org/0000-0002-5493-6424

**Decision letter and Author response**
Decision letter https://doi.org/10.7554/eLife.95830.sa1
Author response https://doi.org/10.7554/eLife.95830.sa2

## Additional files

### Supplementary files
• MDAR checklist
• Supplementary file 1. Fly lines used in the study.
• Supplementary file 2. Antibodies used in the study.
• Supplementary file 3. A full list of Hkb target genes, GO clusters, and the list of genes in each cluster.

### Data availability

All data generated or analyzed during this study are included in the manuscript and supplementary files. *Supplementary file 3* contains all the full list of Hkb target genes, GO clusters, and the list of genes in each cluster.

The following previously published dataset was used:

| Author(s) | Year | Dataset title | Dataset URL | Database and Identifier |
|---|---|---|---|---|
| White KP | 2010 | hkb_ChIPSeq_1 and 2 | https://www.ncbi.nlm.nih.gov/geo/query/acc.cgi?acc=GSM569793 | NCBI Gene Expression Omnibus, GSM569793 |

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
