## [Editor Report]

The important paper provides compelling evidence that the gene hkb controls the development of the salivary gland of *Drosophila* by controlling a cell cycle and cell death. The work extends and corrects previous findings on the role of this transcription factor and will be important for scientists interested in organogenesis and in particular the coordination of cell cycle and cell death.

---

## [Decision Letter]

[Editors' note: this paper was reviewed by Review Commons.]

---

## [Author Response]

We are delighted that all three reviewers were enthusiastic about the work. Their comments and suggestions have improved the paper. The details of the changes we have made in response to each reviewer’s comments are included in blue text below.

Reviewer #1 (Evidence, reproducibility and clarity (Required)):This manuscript by Matthew and colleagues reported the role of Hkb, a transcription factor, in coordinating the cell cycle and morphogenesis during embryonic salivary gland (SG) development. Although the phenotype of hkb SG has been described, the potential mechanism for how Hkb regulates the salivary gland invagination position was still unclear. The authors of this manuscript generated strong hkb mutant lines and confirmed the SG invagination defect. Also, the expression of Hkb in the SG was examined by using hkb knock-in reporter lines. Cell number analysis indicated that the SG cell number was reduced in the hkb mutant. Interestingly, the authors found that both cell proliferation (pH3-positive cells) and cell death (Dcp-1-positive cells) were increased in the hkb mutant SG. Regarding the mechanism, the authors found that Hkb may regulate cell cycle-related genes based on available ChIP-seq data. Overall, this work illuminates the role of Hkb during SG formation.Major comments:1. In lines 337-338, 'we found that the mispositioning of the SG was due to the malformation of the duct', this conclusion lacks sufficient evidence to support it. Does Hkb express in the duct cells? Can SG defects be rescued if duct defect is rescued?

Thank you for the comment. First, we would like to clarify that many widely used SG markers, such as Fkh and CrebA, are specifically expressed in secretory cells from mid-stage 11 onward. When the SG is specified at stage 10, both secretory and duct cells express Fkh and CrebA, but soon thereafter they are restricted to the secretory cells. Only a few markers are expressed specifically in duct cells, such as Dead Ringer. In most of our analyses we have used CrebA signals to mark the SG, We have indicated in the text that CrebA is a nuclear marker for SG secretory cells when CrebA is first introduced.

As shown in Figure 1—figure supplement 5, a salivary duct is composed of one common and two individual ducts and forms in the head region of the embryo; the secretory portion of the salivary gland forms at the end of the individual duct. Our original statement that 'we found that the mispositioning of the SG was due to the malformation of the duct' was based on the mispositioning of the SG in the anterior head region, which is associated with the significantly shortened duct phenotype with fewer duct cells (a few duct cells show DCP-1 signals, suggesting cell death), the smaller apical domain of duct cells, and more cells around the circumference of the duct (Figure 1—figure supplement 5).

Using the Hkb^WT^-mCherry signals, we found a very transient expression of Hkb in some duct cells at early stage 11 of embryogenesis (Figure 1—figure supplement 5). While this suggests a potential role for Hkb in the duct, due to the lack of a driver that only expresses a transgene in the duct, we could not perform a rescue experiment to test whether the SG defect can be rescued when the duct defect is rescued. Therefore, we have modified our original statement to: ‘the mispositioning of the SG may be due in part to the defects of the duct, likely resulting from a combination of reduced cell number, aberrant apical domain size, and defective cell intercalation of duct cells’, implying that the defect we see in SG positioning may have multiple contributing factors.

2. Hkb expression was restricted to the distal region of SG at stage 12, and no Hkb expression was detected after Stage 13 (Figure S5), how does Hkb determine the proper SG formation?

We would like to clarify the Hkb expression in the SG. Using Hkb^WT^-mCh signals, we show that Hkb expression is detected in the SG from stages 10-13 (Figure 1—figure supplement 2). At stages 10 and 11, strong Hkb^WT^-mCh signals are detected in the SG. While higher levels of Hkb are detected in the posterior region of the placode at stage 10, all SG cells show strong Hkb expression at stage 11. At stages 12 and 13, all SG cells express weak Hkb^WT^-mCh signals.

Our data show that Hkb primarily regulates the proper cell cycle progression and SG cell survival during stages 11-13 to ensure proper SG formation, which correlates with the timing of Hkb expression. In addition, as shown in Sanchez-Corrales et al., 2021 and some of the ChIP-seq targets of Hkb identified in our study, Hkb also plays a role in the SG beginning at the onset of SG morphogenesis at stage 11 with the potential function of regulating the actomyosin-mediated contractility of SG cells undergoing apical constriction. In our rescue experiment, we show that overexpression of Hkb in the SG using fkh-Gal4 was sufficient to rescue most of its loss-of-function defects (Figure 2; Figure 2—figure supplement 2), suggesting the tissue-autonomous role of Hkb in SG morphogenesis.

Reviewer #1 (Significance (Required)):Altogether this is an interesting article that contributes significantly to the function of Hkb in organ formation.

Thank you for the valuable and thoughtful comments.

Reviewer #2 (Evidence, reproducibility and clarity (Required)):Matthew provide an in depth analysis of the role of the SP1/KLF-like Huckebein (hkb) transcription factor during embryonic development of the *Drosophila* salivary gland, which provides an excellent representative tissue for understanding tubulogenesis and how cell shapes and movements are coordinated with cell cycle progression during organ formation. Interestingly, hkb function represses both entry into mitosis and apoptosis, resulting in failures of proper salivary gland invagination and elongation. This work provides an interesting example of how coordinating cell cycle progression with morphogenesis is developmentally essentially. The ample data describing the observations is quite strong and well described. However, some of the interpretations of these data, particularly with respect to cell cycle control, do not fit what is generally known from previous work, and thus the final conclusions and summary model need to be supported by more extensive evidence or phrased to consider alternative hypotheses. The critique provides more detail on this point as well as an alternative way for the authors to consider their results.Reviewer #2 (Significance (Required)):The FUCCI system is great, but it only measures S phase indirectly. In the hkb mutants it seems that the E2f1 reporter never really goes away entirely, as typically happens during S phase, and rather seems variable at stage 13. Thus, it's unclear whether the SG cells of hkb mutants enter the first endo S phase at all, or whether they enter a weak S phase that might not eliminate all of the E2f1. The authors really should use EdU labeling in addition in the hkb mutants to assess DNA replication directly. This labeling in embryos can be a little tricky to do, but the results will be much more definitive.

Thank you for the nice suggestion. As suggested, we used EdU labeling to confirm our data with the FUCCI system and added the new data to Figure 5. Consistent with the FUCCI data, we observed different intensities of EdU signals in *hkb* mutants; some SGs showed no EdU signals at all, suggesting that no cells are entering S phase, while others had a small number of cells with weaker EdU signals compared to WT SGs of similar stages, suggesting that they may be entering a weak S phase. A brief discussion of this has been added to Discussion.

In addition, after analyzing more Fly-FUCCI data for late-stage SGs in combination with EdU labeling, we found that the second endocycle in WT SGs at stage 15/16 also appears to be a progression from distal to proximal rather than a random, asynchronous one. The new data have been described (Figure 5R-R’’, T).

Regarding the data, it seems odd to me that the Hkb rescue, which isn't very strong given that WT has NO mitotic cells, is called not significantly different from WT whereas the p35 rescue is, especially because the Hkb rescue and the p35 rescue are not called significantly different from one another. Seems like the Hkb rescue didn't work very well at all here. Thus, the conclusion of compensatory proliferation not being involved is rather weak and should just be taken out or softened substantially.

The original Hkb rescue experiment performed at 25℃ showed a partial rescue of the phenotypes. Even under this condition, we observed a significant reduction in the number of dividing cells, which was further reduced when we performed this rescue assay at 29℃. Since Hkb expression at 29℃ almost completely rescued all the *hkb* mutant phenotypes, we replaced the rescue data with those at 29℃ (Figure 2G’’, P; Figure 4M; Figure 2—figure supplement 2). On the other hand, p35 overexpression, which was shown to work well at 25℃ based on the significantly reduced number of dying cells, did not reduce the number of dividing cells; quite a few SG cells still express PH3 signals in this condition, leading to our conclusion that compensatory cell proliferation may not be at play in the *hkb* mutant SG placode. We have included a statistical analysis to show that the number of dividing cells in the Hkb and p35 overexpression assays are significantly different from each other (Figure 4M). However, since we cannot completely rule out the possibility that compensatory proliferation by a few remaining dying cells contributes to the population of dividing cells in the mutant SG, we have softened the conclusion.

The Cdk1 activating phosphatase string (stg) controls entry into embryonic mitosis in *Drosophila*, but the authors never discuss stg or do any experiments. A stg knockdown in the hkb mutant SG might rescue all the phenotypes much better than any of the other knockdowns performed. Also, might stg be a hkb target gene, and might repression of stg by hkb account for all the phenotypes? I say this because this statement of data interpretation in the Discussion line 569 is likely false: "We propose that Hkb promotes SG cell entry into the endocycle through its repression of E2F1 and CycE expression." False because E2f1 and CycE are REQUIRED for entry into S phase and endocycle progression (lots of previous publications). Thus, this conclusion doesn't make sense to me. Rather, perhaps hkb is only repressing string, or at least this is the important control, and when this doesn't happen the E2F1/CycE driven S phase (i.e. the first endo S in stage 12/13 in WT) results in entry into mitosis rather than a pause in G2 prior to the next asynchronous endo S phase in stage 15. I would argue that this could account for all the phenotypes and that the data showing elevated E2f1 and CycE transcripts (which is marginally convincing at best unfortunately) are a red herring.

Thank you for the valuable comments. We also initially thought about the possibility of *stg* upregulation/derepression in *hkb* mutants but did not discuss it in the original manuscript because *stg* was not a Hkb ChIP-seq target (Supplementary file 3). In the revised manuscript, we added an explicit statement in the text that *stg* is not a ChIP-seq target. Furthermore, using fluorescence in situ hybridization, we confirmed that *stg* expression is NOT upregulated in *hkb* mutants and included these new data in Figure 6—figure supplement 1. *stg* expression in the SG is very low compared to neighboring tissues both in WT and *hkb* mutants, and quantification showed that *stg* expression levels were not significantly different in the *hkb* mutant SG compared to WT. We therefore rule out the possibility that the *hkb* mutant phenotype is due to *stg* upregulation.

We agree that our previous statement in the Discussion "We propose that Hkb promotes SG cell entry into the endocycle through its repression of E2F1 and CycE expression." did not accurately explain the data. As the reviewer aptly pointed out, E2f1 and CycE are required for entry into S phase and endocycle progression. However, several studies have shown that E2F1 should be repressed at the onset of S phase of the endocycle, whereas CycE repression should occur later in S phase; a process that is also required for the switch back to G phase of the endocycle (Shibutani et al., 2008; Zielke et al., 2011). We have restated that “Hkb promotes SG cell progression through the S phase of the endocycle, at least in part through its repression of E2F1 and CycE expression”.

Finally, could fzr be a target of hkb? Activation of fzr and repression of stg by hkb would synergize to drive the cells into an endoreplication program.

Fzr is not a Hkb ChIP-seq target, and we have explicitly stated this in the text. Our fluorescence in situ hybridization data showed that *fzr* mRNA is upregulated in both WT and *hkb* mutant SGs (data not shown), and it is inconclusive whether *fzr* mRNA levels are different in *hkb* mutants compared to WT. We cannot rule out that *fzr* levels are slightly reduced in *hkb* mutant SGs.

In sum, the authors provide a lot of excellent data but I feel should rethink their interpretations a little bit to build a model that might match better what is already known regarding cell cycle regulation.

Thank you for the thoughtful comments. We have carefully revised our interpretations to explain the role of Hkb in cell cycle regulation during *Drosophila* salivary gland morphogenesis. Some of our findings were indeed surprising, including the lack of a clear role for *stg* in this context. We believe that our findings provide a compelling illustration of the sophisticated regulatory mechanisms that control the cell cycle during organ formation.

Reviewer #3 (Evidence, reproducibility and clarity (Required)):The study by Matthew et al. provides a meticulous characterization of their newly generated null hkb alleles including mCherry tag to visualize Hkb localization during *Drosophila* salivary gland development in embryogenesis. In contrast to a recent report, the authors discover that Hkb primary function is to regulate cell cycle and cell death genes during tissue morphogenesis. This study finds that genetic loss of hkb primarily results in enhanced cell death (DCP1+) and increased mitotic division (P-HH3+) with a more minor contribution from apical constriction. To support this conclusion the authors show using FISH that genetic loss of hkb results in upregulation of cell death gene: reaper and cell cycle genes: E2F1, CycE, and CycA. However, the FISH staining is not quantified and in some cases the staining does not seem significantly changed weakening the author conclusions. Lastly, the authors show that the observed morphological defect in salivary gland development can be partially rescued by combining the hkb null mutant with the knockdown of either CycE, E2F1 or overexpression of Fizzy-related (an E3 ligase that inhibits mitotic cyclins). Therefore, the study has revealed a novel function for Hkb in regulation of SG cell cycle timing during development.Major comments:1) The FISH assays need to be quantified to determine whether there is or not a significant change in gene expression (Figures 6 and S12). The change reported in text and a major conclusion of this study is not immediately apparent for E2F1 and CycE.

We have included quantifications of mRNA levels for *rpr*, *E2f1*, and *CycE* for stage 11 SGs in Figure 6, which show that there is a significant increase in the transcript levels of these genes in *hkb* mutants compared to WT, supporting our conclusion. We measured the mean gray value of the mRNA signal intensity in each SG placode and performed a careful background subtraction using the same regions of the embryo that showed low signal intensity in WT and *hkb* mutant conditions. Statistical comparison was performed in GraphPad Prism using Student’s t-test with Welch's correction.

2) A negative control for the hkb rescue assay would be very helpful to further support the authors conclusion that the primary function of Hkb is to repress cell cycle genes and not via the regulation of apical constriction originally reported. This could be accomplished by expressing or knocking down regulators of apical constriction as reported in this study (i.e. Rok and/or myoII mutants).

We would like to point out that although our data suggest that the primary function of Hkb is to regulate key cell cycle genes, we are not arguing that Hkb does not have a role in regulating apical constriction. As we discussed in the Discussion, we propose that in addition to regulating key cell cycle genes, Hkb has a role in regulating apical constriction by regulating cytoskeletal networks. ChIP-seq targets identified key regulators of actomyosin networks, which also supports a model for Hkb in regulating actin polymerization, a process crucial for actomyosin-mediated contractility during apical constriction, and intracellular trafficking components, another important process linked to actomyosin contractility during apical constriction (Le and Chung, 2021). Moreover, SGs with blocked apoptosis still exhibited apical constriction defects (quantification of apical areas shown in a new Figure 3G, G’), supporting our model that Hkb regulates apical constriction.

3) Also, it would further strengthen the authors conclusion that Hkb inhibition of both cell death and cycle genes is necessary for SG development if they could assess whether co-expression p35 and E2F1 RNAi in a hkb null mutant enhances the rescue of SG development.

Thank you for the nice suggestion. As suggested, we performed an assay to simultaneously inhibit both cell death and cell division by overexpressing p35 and Fzr in the *hkb* mutant background (Figure 6—figure supplement 4). We chose to overexpress Fzr rather than knock down *E2f1*, as Fzr, when overexpressed in tissues, acts as a repressor of mitosis via its repression of the mitotic cyclins, shunting these tissues into the endoreplication cycle. We observed an enhancement of the rescue phenotype compared to inhibition of cell death or cell division alone; a more significant decrease in the number of dying and dividing cells, as well as an increase in the number of elongated SGs.

Reviewer #3 (Significance (Required)):This study extends our knowledge of how a conserved transcription factor, Hkb, functions in tissue morphogenesis during animal development using the salivary gland of the fruit fly as a model. This study will be of particular importance to developmental biologist who seek to understand the mechanistic regulation of tubule formation and tissue morphogenesis. The study's conclusion that Hkb functions primarily as a regulator of cell cycle and cell death will warrant further investigation into its role in tissue morphogenesis in other tissues in the fruit fly as well as in mammalian development. Therefore, this study will be of broad interest to basic research community.My field of expertise is cell biology and genetics with a focus on polyploidy.

Thank you for the valuable and thoughtful comments.